# In-plane staging in lithium-ion intercalation of bilayer graphene

**Thomas Astles[1,3], James G. McHugh [1,2,3], Rui Zhang [1,3], Qian Guo[1,2], Madeleine Howe[1], Zefei Wu[1,2], Kornelia Indykiewicz [1,2], Alex Summerfield [2], Zachary A. H. Goodwin[1,2], Sergey Slizovskiy [1,2], Daniil Domaretskiy [1], Andre K. Geim[1,2], Vladimir Falko [1,2] ✉ & Irina V. Grigorieva [1,2] ✉**

The ongoing efforts to optimize rechargeable Li-ion batteries led to the interest in intercalation of nanoscale layered compounds, including bilayer graphene. Its lithium intercalation has been demonstrated recently but the mechanisms underpinning the storage capacity remain poorly understood. Here, using magnetotransport measurements, we report *in-operando* intercalation dynamics of bilayer graphene. Unexpectedly, we find four distinct intercalation stages that correspond to well-defined Li-ion densities. Transitions between the stages occur rapidly (within 1 sec) over the entire device area. We refer to these stages as 'in-plane', with no in-plane analogues in bulk graphite. The fully intercalated bilayers represent a stoichiometric compound $C_{14}LiC_{14}$ with a Li density of $\sim 2.7 \cdot 10^{14}\,cm^{-2}$, notably lower than fully intercalated graphite. Combining the experimental findings and DFT calculations, we show that the critical step in bilayer intercalation is a transition from AB to AA stacking which occurs at a density of $\sim 0.9 \cdot 10^{14}\,cm^{-2}$. Our findings reveal the mechanism and limits for electrochemical intercalation of bilayer graphene and suggest possible avenues for increasing the Li storage capacity.

Graphite as an anode material for Li-ion batteries has been studied extensively for many decades and is currently being employed as an essential component in rechargeable batteries[1–4]. Often used in the form of graphite powder baked onto copper foils, it offers many advantages for battery anodes, such as chemical inertness, reversible intercalation, good cyclability, and relatively low costs. In an effort to further enhance the performance of graphite-based anodes, recent research focused on replacing graphite with few-layer graphene and attempting to establish the factors that govern the intercalation of Li ions, particularly into bilayer graphene (BLG) that presents the elementary building block of the AB-stacked (Bernal) graphite[5–17]. Promisingly, Li diffusion rates in BLG are found to be two orders of magnitude higher than in bulk graphite[5], which may provide much faster charging-discharging. On the other hand, the storage capacity for BLG – Li-ion density achievable by electrochemical intercalation –

has so far been disappointingly low ($n_{Li} \lesssim 2 \times 10^{14}\,cm^{-2,5,9,10}$), that is, 3 times lower than that achieved for stage-I intercalation of bulk graphite ($LiC_6$)[18–21]. The only exception is a recent study where intercalation was monitored in a transmission electron microscope (TEM), which reported the formation of multilayered metallic lithium[7]. In the latter case, however, the high-energy electron beam might interfere with the electrochemical process by, for example, contributing to Li ions reduction. The question remains open whether Li-ion storage above $\sim 2 \times 10^{14}\,cm^{-2}$ can be achieved in operando, using the standard electrochemistry most relevant for battery technologies.

Understanding the limits for the use of bilayer and few-layer graphene for Li-ion storage requires knowledge of mechanisms governing Li intercalation, including any structural changes, arrangements of Li ions relative to the underlying graphene lattice, and effects of the disorder. While the above factors have been the focus of many

[1]Department of Physics and Astronomy, University of Manchester, Manchester, UK. [2]National Graphene Institute, University of Manchester, Manchester, UK. [3]These authors contributed equally: Thomas Astles, James G. McHugh, Rui Zhang. ✉e-mail: Vladimir.Falko@manchester.ac.uk; Irina.V.Grigorieva@manchester.ac.uk

computational studies[11,13–17,22], experimental evidence remains scarce. Among other outstanding questions is the possibility of in-plane staging, that is, of a sequence of preferential Li configurations replacing each other during intercalation, by analogy with alkali ion intercalation of bulk graphite[23–27]. Such staging has been suggested to explain certain features of Li intercalation of a 'bilayer foam', a bulk system consisting predominantly (but not exclusively) of graphene bilayers[8]. However, no signatures of in-plane staging were reported in several other studies of isolated BLG[5,6,9], implying that the features observed in ref. 8 could have a different origin (e.g., staging in graphene multilayers present in that system, or elastic strain[18,28]).

To shed light on the above uncertainties, in this work, we study Li intercalation of BLG using an on-chip electrochemical cell where the entry and exit of Li ions are examined via changes in resistivity and carrier density of graphene. The intercalation process is continuously monitored over many intercalation-deintercalation cycles, while the rate of the intercalation reaction is kept sufficiently slow to resolve individual stages. We observe four distinct plateaus in graphene resistivity, corresponding to distinct densities of intercalated lithium, $n_{Li}$. We refer to these as in-plane stages (I–IV) so as not to confuse the effect with the well-known staging in bulk graphite, where Li ions fill interlayer spaces to the full capacity in one step. Lithium intercalation in our device setup is further verified by *in operando* Raman spectroscopy. Our measurements of Li-ion densities for stages III and IV allow us to identify them as $C_{18}LiC_{18}$ and $C_{14}LiC_{14}$, where Li ions attain hexagonal arrangements commensurable with the underlying graphene lattice. Assuming similar commensurability that minimizes Coulomb interactions, low-density stages II and I are identified as $C_{38}LiC_{38}$ and $C_{42}LiC_{42}$. Combining the experimental findings and DFT calculations, we show that $C_{14}LiC_{14}$ corresponds to the thermodynamic equilibrium for intercalated AA-stacked bilayers. Another observed stage ($C_{18}LiC_{18}$) is close to this equilibrium configuration, whereas stages I and II at much lower $n_{Li}$ are attributed to metastable states within the original AB stacking. Transitions between the stages occur rapidly (typically within 1 sec) over the entire device area. Our DFT analysis suggests that the transition between the two pairs of in-plane staging is accompanied by changing the BLG structure from AB to AA stacking, which occurs at a threshold Li ion density in the AB bilayer and is facilitated by the formation of AB/BA boundaries during intercalation-deintercalation cycles.

## Results

### Setup of the on-chip electrochemical cell

A schematic of the experimental setup and an optical image of one of our devices are shown in Fig. 1a, b. BLG was mechanically exfoliated onto a Si/SiO$_2$ substrate and shaped into a Hall bar, with a few μm of its edge being exposed to solid polymer electrolyte LiTFSI-PEO ("Methods"). A gate voltage $V_g$ between the Pt counter electrode and BLG provided a controlled driving force for electrochemical intercalation. To protect the device from degradation, most of the Hall bar and Au contacts were covered with a passivating layer of SU-8 resist as shown in Fig. 1b and Supplementary Fig. 1a. This design ensured that Li ions from the electrolyte could enter BLG's interlayer space (gallery) only through the exposed edge. All measurements were carried out in the inert environment of an Ar-filled glovebox with < 0.5 ppm oxygen and moisture levels to prevent degradation of the electrolyte. The temperature was kept at $330 \pm 2$ K to ensure sufficient electrolyte conductivity ("Methods" and Supplementary Fig. 1c, d). The large thickness of the SU-8 layer ($\sim 5$ μm) prevented ionic gating from the surrounding electrolyte that could, in principle, contribute to changes in the carrier density and resistance of the bilayer. Control experiments on monolayer graphene in a similar setup confirmed that Li ions did not intercalate between graphene and SU-8; intercalation occurred only in the gallery of the bilayer, in agreement with previous studies[5,9].

Intercalation and deintercalation were monitored as a function of time $t$ at 1 s intervals via measurements of graphene's resistivity $\rho_{xx}$ and Hall voltage $V_{xy}$ using the standard lock-in technique. Employing different pairs of contacts allowed us to probe the uniformity of intercalation over the $\sim 20$ μm length of our bilayer devices (Fig. 1b). If we swept $V_g$, a sharp peak appeared in $\rho_{xx}$ and a spike-like feature in $V_{xy}$ at a critical value of about $- 3$ V (Fig. 1c and Supplementary Fig. 2), which indicated that our initially p-doped devices (Supplementary Methods 1.1) changed their doping polarity and the Fermi level passed through the neutrality point as the result of Li ions entering BLG[5,9]. In measurements of $\rho_{xx}(t)$ and $V_{xy}(t)$, we mostly used $V_g = -7$ V. Smaller $V_g$ required longer times to achieve the same intercalation level but did not affect the discussed results, which is consistent with the known behavior for Li intercalation of graphite[29]. Further details explaining the working of our electrochemical cell are provided in Supplementary Methods 1.1. Below, we focus on the behavior exhibited by one of our devices that was studied in greater detail (device A), and another device B is described in Supplementary Methods 1.1 and Supplementary Fig. 4. The reported in-plane staging was observed in all five studied BLG devices, with not only qualitatively but also quantitatively the same characteristics (Supplementary Fig. 6).

### Monitoring intercalation-deintercalation cycles via evolution of the bilayer graphene resistance

Typical evolution of BLG's resistance over consecutive intercalation cycles is shown in Fig. 1d with representative cycles shown separately in Fig. 1e. The most notable feature here is the presence of resistance plateaus with well-defined $\rho_{xx}$ values that persisted over extended time intervals up to 20 min and, importantly, reappeared in different cycles and for all the studied devices. Only the time span of the plateaus varied from cycle to cycle (Fig. 2c). Transitions between the plateaus occurred typically within 1 sec (our time resolution) and simultaneously across the whole device, see Fig. 2a for a typical example. Studying different cycles and devices, we found that $\rho_{xx}$ for the four observed plateaus was on average $\sim 248, 219, 142,$ and $134$ Ω with a standard deviation of $\pm 3\%$ (Fig. 1d, e and Supplementary Fig. 4a, b). Plateaus in $\rho_{xx}$ were accompanied by plateaus in $V_{xy}$ measured simultaneously (Supplementary Fig. 3b). To determine the Hall resistivity $\rho_{xy}$ at the plateaus, we reversed the magnetic field (switching it between $\pm 330$ mT by rotating a permanent magnet around the intercalation/measurement setup within the glovebox). This allowed us to avoid spurious offsets in $V_{xy}$ which sometimes appeared from a $\rho_{xx}$ contribution. From the $\rho_{xy}$ value, we found the electron density $n$ induced in BLG and then estimated $n_{Li}$ using the known charge transfer of $\sim 0.9e$ from each Li-ion to graphene[11,14,15] ($e$ is the electron charge), see Supplementary Methods 1.1 for details. The Li-ion density for the observed plateaus was $\sim 0.9, 1.0, 2.1,$ and $2.7 \times 10^{14}$ cm$^{-2}$ (Fig. 3a). In the following, we refer to these four distinct states of BLG intercalation as in-plane stages I to IV, respectively.

Achieving reproducible in-plane staging required several cycles of intercalation/deintercalation. During the initial few cycles, $\rho_{xx}$ and $V_{xy}$ fluctuated with time without acquiring any specific values (Supplementary Figs. 2c, 3a) despite $V_g$ being kept constant and applied for several hours. Remarkably, even in this weakly-doped state the observed fluctuations accurately reproduced at all voltage contacts (Supplementary Fig. 2c, e), which showed that Li-ion doping occurred practically simultaneously over the entire 20 μm long device and Li ions rearranged themselves into different configurations within 1 sec. This also meant that no macroscopic domains were formed, in contrast to, e.g., Li intercalation of bulk Li$_x$CoO$_2$[30]. After the first couple of cycles, stage I with its $\rho_{xx} \approx 248$ Ω and $n_{Li} \approx 0.9 \times 10^{14}$ cm$^{-2}$ clearly developed, persisting for a long time (inset of Fig. 1d and Supplementary Fig. 3a). In the 4th or 5th cycle and after long (> 2 h) exposures to $V_g = -7$V, we also started to observe stage IV (lowest $\rho_{xx} \approx 134$ Ω and highest $n_{Li} \approx 2.7 \times 10^{14}$ cm$^{-2}$; inset of Fig. 1d and left panel of Fig. 1e). The

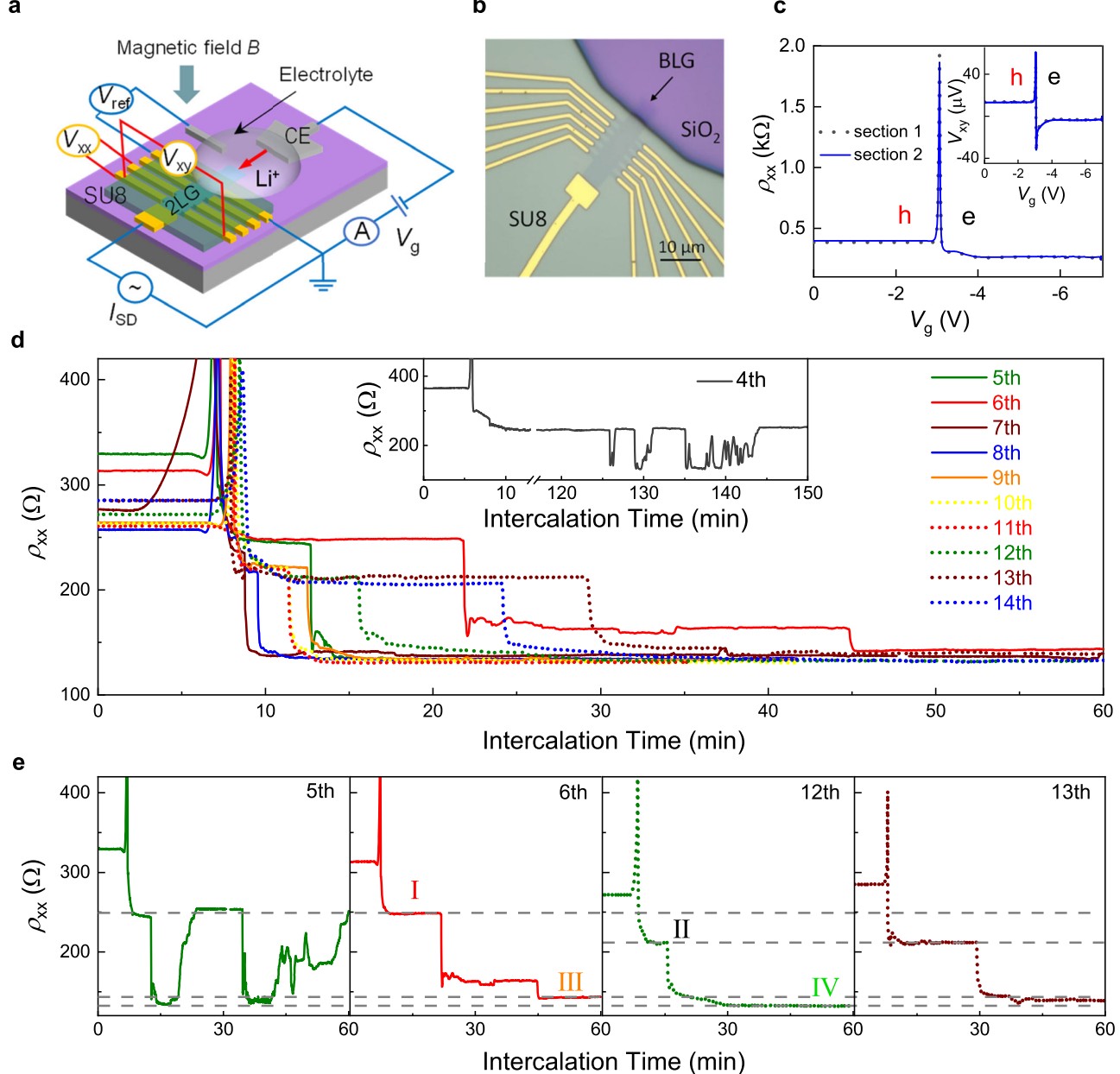

**Fig. 1 | Device design and the evolution of the bilayer graphene's resistance over multiple intercalation cycles. a** Schematic of the experimental setup. Bilayer graphene (2LG) is mechanically exfoliated onto a Si/SiO₂ substrate and shaped into a Hall bar, with a few µm of its edge exposed to a solid polymer electrolyte. A gate voltage $V_g$ between the Pt counter electrode and BLG provides a controlled driving force for electrochemical intercalation. To protect the device from degradation, most of the Hall bar and Au contacts are covered with a passivating layer of SU-8 resist. Longitudinal and Hall voltages, $V_{xx}$ and $V_{xy}$ respectively, are measured continuously during intercalation/deintercalation using different pairs of contacts, with the current $I_{SD}$ applied through the BLG. In addition, a reference potential $V_{ref}$, measured between the graphene bilayer and a pseudo-reference electrode, and the current through the electrolyte is used to monitor the intercalation. **b** Optical image of a typical device (device A). **c** Longitudinal resistance (main panel) and Hall voltage (inset) vs the gate voltage. Solid blue and dotted black curves, corresponding to different sections of the device, overlap. **d** Evolution of BLG resistance $\rho_{xx}$ during the first hour in consecutive intercalation cycles (typically measured at 1 s intervals; color-coded). The inset shows multiple jumps in $\rho_{xx}$ observed in the 4th intercalation cycle between resistance values corresponding to transitions between stages I and IV (see text). **e** Detailed time evolution for several representative intercalation cycles; the label in each panel gives the cycle number. Dashed horizontal lines indicate the average values of $\rho_{xx}$ for the found four stages (see text). Infrequently, we also saw plateaus that were not reproduced in any other cycle (see, e.g., the plateau at about 160 Ω in the 6th cycle). Source data are provided as a Source Data file.

latter plots show sharp jumps between the resistance plateaus corresponding to stages I and IV, but the high-$n_{Li}$ states persisted no longer than a few minutes at a time. Only after further cycles did all four in-plane intercalation stages become established and recur for extended periods of time (Fig. 1d, e). Having said that, in later cycles, stage III became poorly defined (transient), gradually transforming into stage IV (Supplementary Fig. 3c and Fig. 1e). In addition, in these later cycles

our devices tended to bypass stage I, immediately entering stage II and then making a two-/three- fold jump (in $\rho_{xx}$ and $n_{Li}$, respectively) into the most-doped state (stage IV). The observed 'initial softening or training' of graphite-based systems during intercalation is well known in the literature and attributed to the gradual expansion of the inter-layer space during initial intercalation cycles[12,18,21]. Let us also emphasize that no further increase in Li density beyond $n_{Li} \approx 2.7 \times 10^{14}$ cm⁻² for

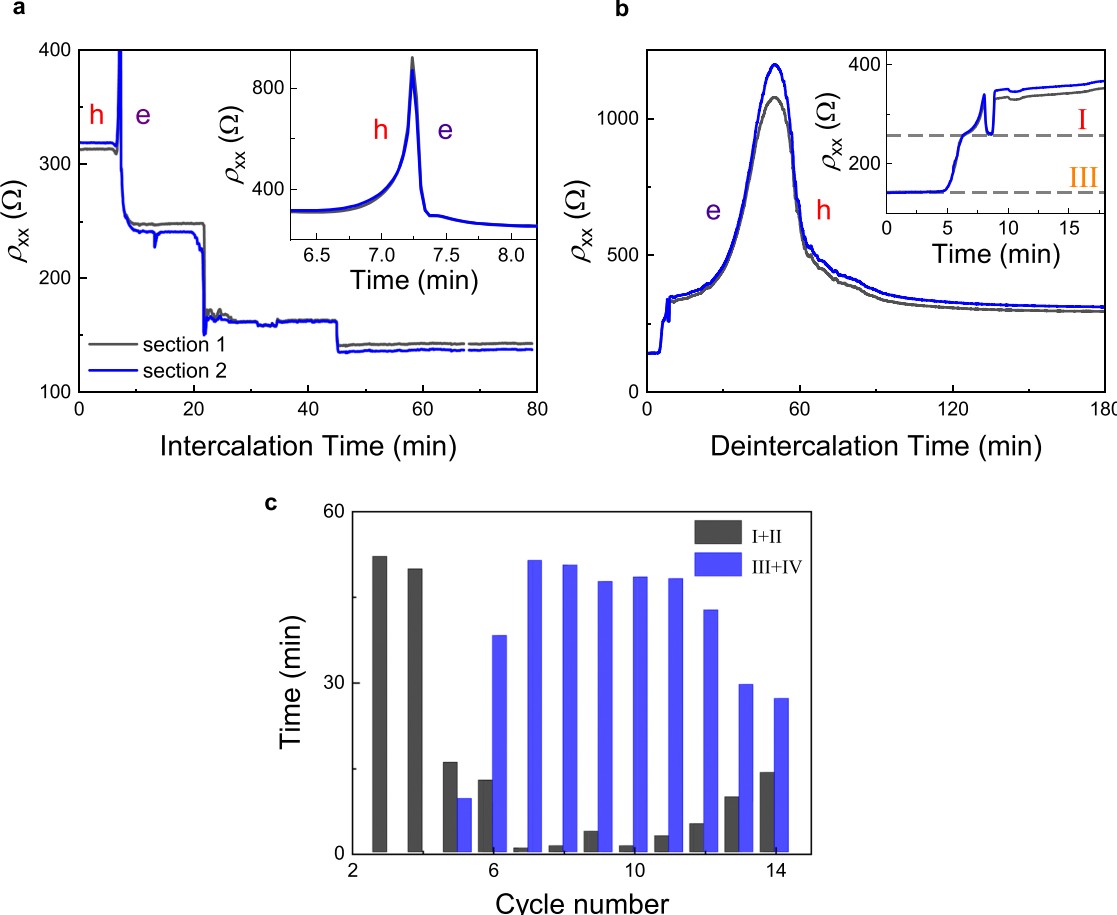

**Fig. 2 | Simultaneous changes and kinetics of intercalation stages.**
**a**, **b** Representative time evolution of the BLG resistance $\rho_{xx}$ for two sections of device A during intercalation (**a**) and deintercalation (**b**); 6th intercalation-deintercalation cycle. Inset in (**a**): zoom-in of the resistance peak corresponding to the change from p- to n-doping as Li ions enter BLG. The inset in (**b**) shows typical kinks in $\rho_{xx}$ during deintercalation. Dashed lines in (**b**) correspond to 258 $\Omega$ and 143 $\Omega$, resistance values characteristic of stage I and stage III intercalation, respectively. **c** Overall time spent at stages I and II (AB stacking) and stages III and IV (AA stacking) during consecutive 60-minute intercalation cycles. Source data are provided as a Source Data file.

stage IV could be detected in any of our devices, irrespective of the driving potential, intercalation time, and other conditions. This doping, is in fact, comparable to or higher than those reported previously for Li intercalation of BLG[5,9]. This suggests that stage IV represents the intrinsic capacity of BLG for the standard electrochemical intercalation with Li.

As for deintercalation, in contrast to the fast entry of Li ions into the interlayer space, the return to a deintercalated state (after $V_g$ was set back to zero) was much slower and became progressively slower still with each cycle (Fig. 4 and Supplementary Fig. 5). Kinks and small steps indicating the in-plane staging were also observed on $\rho_{xx}(t)$ curves during deintercalation (insets of Figs. 2b and 4) but they were much less pronounced and lasting than those for intercalation half-cycles. Furthermore, unlike the well-defined resistance state reached after each full intercalation to stage IV, $\rho_{xx}$ after deintercalation varied considerably and usually decreased with repeated cycling (see Fig. 1d at $t = 0$) whereas the final carrier density (p-doping) reached after deintercalation remained approximately the same. This is unexpected because disorder induced by intercalation/deintercalation cycles should generally increase the resistivity. Indeed, another notable feature of deintercalation is that the resistance peak at the transition from n- to p- doping became progressively broader and smaller with repeated cycling (Supplementary Fig. 5), which indicated increasing inhomogeneity of deintercalated BLG[31], as expected. Only after many intercalation cycles (typically >10), disorder started playing a critical

role in the measured characteristics. Eventually, after 15–20 cycles, intercalation becomes progressively less effective, and the maximum Li concentrations of $\sim 2.7 \times 10^{14}$ cm$^{-2}$ could no longer be achieved even after many hours (leading to $n_{Li} \ll 2 \times 10^{14}$ cm$^{-2}$).

## Monitoring intercalation by Raman spectroscopy

As an alternative way to monitor intercalation, we used in situ Raman spectroelectrochemistry[32–34] (details in Supplementary Methods 1.2). Figure 5 shows the evolution of the Raman G and 2D peaks of our bilayer graphene as a function of the applied gate voltage $V_g$. The G peak position shifted sharply by $\sim 8$ cm$^{-1}$ as soon as the gate voltage exceeded the threshold value (3.8 V for this device) while the longitudinal resistance (measured simultaneously) went through a sharp peak, indicating intercalation. Simultaneously, the 2D peak intensity $I(2D)$ became strongly suppressed, with $I(2D)/I(G)$ ratio changing from $\sim 0.8$ before intercalation to $\sim 0.5$ after. This behavior indicates strong electron doping[35,36]. Both peaks returned to the initial position and intensity after deintercalation. The spectra in Fig. 5 were collected during the 2nd intercalation cycle, and similar curves obtained for the 3rd cycle, with the Li ion density reaching stage I/II (Fig. 3a), or carrier density in the bilayer $n \sim (8 - 9) \cdot 10^{13}$ cm$^{-2}$. As expected, both G and 2D peaks remain visible for this doping, very similar to the case of stage-II Li intercalation of graphite, where doping was even higher[37]. Although further cycling led to higher electron density, it was not possible to quantify this evolution using Raman spectra because of the visible

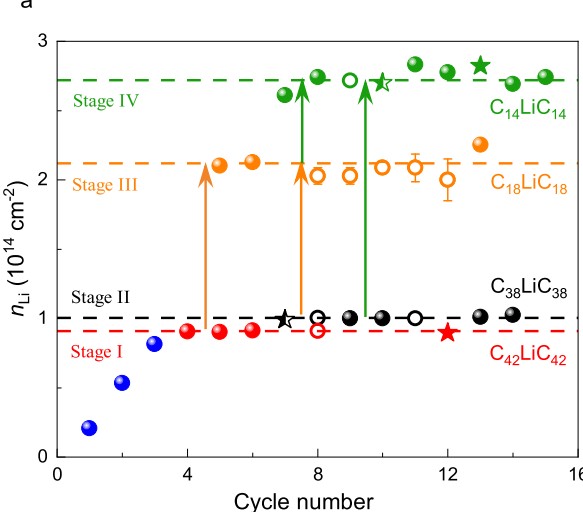

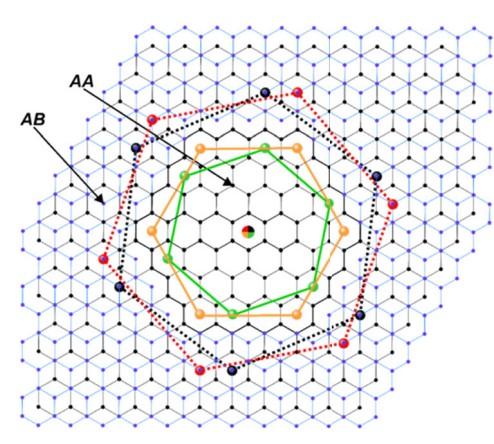

**Fig. 3 | Stages of intercalation. a** Li-ion densities $n_{Li}$ found from Hall measurements (filled symbols). Open symbols show $n_{Li}$ evaluated from measured $\rho_{xx}$ assuming the electron scattering times remained unchanged for a given intercalation stage (Supplementary Methods 1.1). All circular symbols are for device A. For comparison, star symbols show average $n_{Li}$ for corresponding intercalation stages found for two other devices (devices B and C). Horizontal lines show $n_{Li}$ expected for exact $C_{42}LiC_{42}$, $C_{38}LiC_{38}$, $C_{18}LiC_{18}$ and $C_{14}LiC_{14}$ stoichiometries (Supplementary Methods 1.1). The arrows indicate dominant transitions that depend on the cycle number. Error bars for open symbols are due to uncertainty in determining $n$ to infer $n_{Li}$. Source data are provided as a Source Data file. **b** Proposed relative positions of Li ions for different stages of intercalation, color-coded as in (**a**). Stages I and II correspond to the intercalation of the AB-stacked bilayer, and stages III and IV to the AA bilayer. Li-ion positions for AB stacking correspond to one of the two equivalent configurations and are therefore shown by the dashed lines.

degradation of the electrolyte under the laser beam and the strongly increased noise.

The Raman spectra provided a new insight into the intercalation process: Firstly, no D peak above the noise level was observed either before, during, or after intercalation, suggesting that intercalation did not induce point defects in our BLG, in agreement with our conclusions from resistance measurements (no increase in resistance after deintercalation). It is possible that the D peak was comparable in intensity with the noise level, which would give $I_D/I_G < 0.02$ and a minute possible density of point-like defects of $<5 \cdot 10^9$ cm$^{-2}$ (Supplementary Methods 1.2). Secondly, from the observed shift in the G peak position and the change in $I(2D)/I(G)$ intensity ratio, we can place a lower limit on the electron doping induced by intercalation in our devices. As they were always significantly p-doped in the uninteractalated state ($n \sim -10^{13}$ cm$^{-2}$), the shift in the G peak position by $\sim 8$ cm$^{-1}$ corresponds to electron doping much larger than $10^{13}$ cm$^{-2}$ and larger than observed for electrostatically gated bilayer graphene[36]. Notably, the carrier density $n \sim 5 \cdot 10^{13}$ cm$^{-2}$ achieved in ref. 36 is the highest reported in the literature for the experimental evolution of the G and 2D peaks in BLG but twice lower than for our in-plane stage I/II where the Raman spectra were taken. The data scatter in ref. 36. and the saturation in Raman peak positions at high electron doping does not allow extrapolation to higher densities in order to estimate the doping level in our devices. That said, we note that both the G peak shifts and carrier density $n$ extracted from the Hall resistance in our experiment agree with those reported for Li-ion intercalation of BLG in a similar setup in ref. 9.

### Estimating the density of intercalated Li ions from cathodic current

As another alternative method of quantifying the density of intercalated lithium, for one of the devices (Supplementary Fig. 7), we estimated the amount of Li ions entering during intercalation from a step in the cathodic current between the counter electrode and the bilayer, corresponding to a step in $\rho_{xx}$ (Supplementary Fig. 7b). Integrating the current step gives a charge transfer $\Delta Q \approx 4 \cdot 10^{-10}$ C, i.e., entry of $\Delta Q/e \cdot S \approx 5 \cdot 10^{14}$ cm$^{-2}$ electrons, in good order-of-magnitude agreement with the number of intercalated Li ions corresponding to

the transition from stage I to stage III in this case ($\sim 1.3 \cdot 10^{14}$ cm$^{-2}$, Fig. 3a). Here $e = 1.6 \cdot 10^{-19}$ C is the electron charge and $S \approx 5 \cdot 10^{-6}$ cm$^2$ the area of our $\sim 10$ μm x 50 μm device.

## Discussion

The distinct in-plane staging observed in our experiments seems to disagree with the earlier reports for BLG[5,6,9] and with the behavior known for Li intercalation of bulk graphite. In the latter case, it has been shown that, after first Li islands form between graphene planes, the resulting local strain leads to attraction between neighboring islands, which promotes further intercalation, so that the full capacity of an interlayer gallery is reached as soon as it is opened by first intercalating islands[28] or the system phase-separates into a 3D checkerboard structure[38]. As for the existing theory, computational studies usually start with assuming a change in the stacking of the constituent graphene layers from the AB to AA configuration, which occurs concurrently with Li-ion intercalation[17,22,27]. Such a single structural transition cannot explain the occurrence of the observed four stages.

The only plausible explanation for the four well-defined plateaus in $\rho_{xx}$ and $n_{Li}$ separated by sharp transitions is distinct configurations into which Li ions rearrange themselves at different stages. The constant $\rho_{xx}$ also means that only one phase is present at each plateau: If domains of two different phases were present, e.g., an unintercalated and intercalated one, or domains with different Li-ion densities as in graphite[38], we would observe a smooth evolution of $\rho_{xx}$ as the higher density domains grow in size[39] (due to a gradual increase in the amount of charge transfer from Li to graphene). Distinct ion arrangements in our BLG-Li system are perhaps not surprising because this happens for graphite intercalation where Li ions reside at centers of next-nearest carbon hexagons, creating a hexagonal superlattice. Accordingly, some (at least short-range) order can also be expected for the BLG. Indeed, Li ions should not only occupy their energetically most favorable sites at carbon hexagon's centers but also would tend to be spaced equidistantly, to minimize the electrostatic energy due to Coulomb repulsion[40]. Importantly, Coulomb repulsion between Li ions is greatly enhanced in BLG with respect to bulk graphite because of much-reduced screening in 2D[41,42]. This should favor Li-ion ordering at

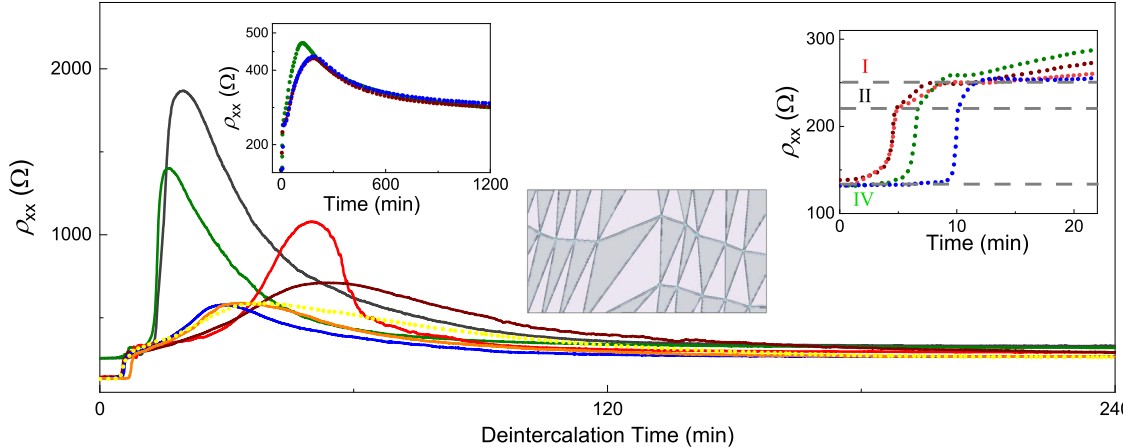

**Fig. 4 | Evolution of BLG resistance during deintercalation.** Main panel: $\rho_{xx}(t)$ during the deintercalation part of consecutive cycles (4th to 14th, as for intercalation in Fig. 1d, same color coding). All deintercalation curves, including the first three cycles, are shown in Supplementary Fig. 5a. The right inset highlights distinct kinks in $\rho_{xx}$ corresponding to transitions between different stages during deintercalation; gray lines correspond to $\rho_{xx}$ = 134, 222, and 250 Ω. The left inset shows evolution of $\rho_{xx}$ during the 12th, 14th, and 15th deintercalation cycles, in which the fully deintercalated state took considerably longer to reach than the time span in the main panel. Middle inset: schematic of AB/BA domains expected to appear in bilayer graphene under cyclical strain. Source data are provided as a Source Data file.

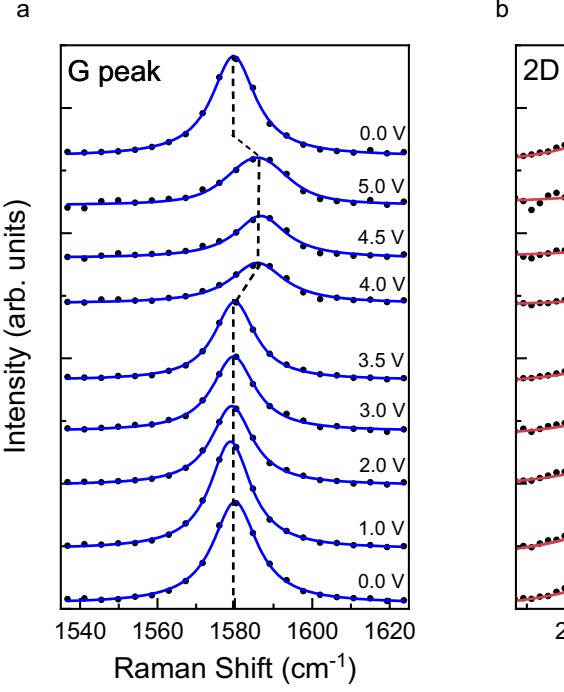

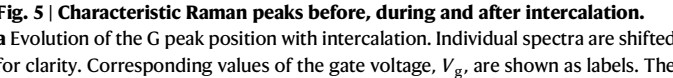

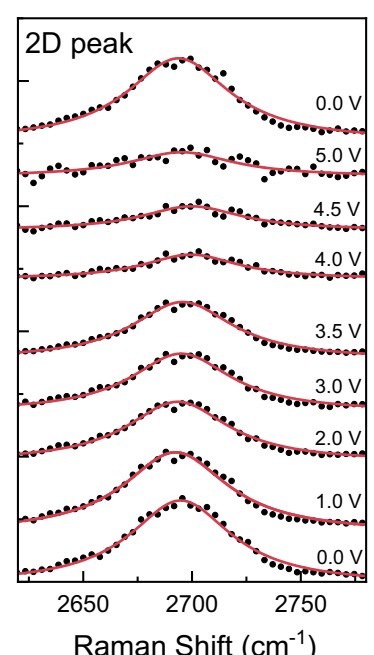

**Fig. 5 | Characteristic Raman peaks before, during and after intercalation. a** Evolution of the G peak position with intercalation. Individual spectra are shifted for clarity. Corresponding values of the gate voltage, $V_g$, are shown as labels. The intercalated state corresponds to $V_g$ = 4.0, 4.5, and 5.0 V. The Black dashed line emphasizes the shift of the G peak in the intercalated state. **b** Same as in (**a**) for the 2D peak. Source data are provided as a Source Data file.

longer distances[43]. The suggested commensurability between Li ions and the carbon lattice (rather than random Li positions) is consistent with the fact that disorder played little role in our experiments. Indeed, the reproducibility of $\rho_{xx}$ within a few % over many intercalation cycles and for different devices indicates that the observed $\rho_{xx}$ values are intrinsic, being determined by electron-phonon scattering, rather than by disorder induced by Li doping. This conclusion about dominant electron-phonon scattering for our Li-intercalated BLG (rather than other scattering mechanisms) agrees with the previous report for heavily doped monolayer graphene which showed that its resistivity at room temperature was phonon-limited[44]. The monolayers exhibited $\rho_{xx}$ of ~100 Ohm at similar doping (~$10^{14}$ cm$^{-2}$), which is a factor of 2 lower than $\rho_{xx}$ in our experiments. However, electron-phonon scattering in BLG should be stronger than in monolayer graphene because of a contribution from shear phonons representing local lateral displacements of the two layers[45] leading to a larger phonon-limited $\rho_{xx}$ compared to the monolayer, as indeed observed for our intercalated BLG.

Assuming commensurability between Li ions and the graphene lattice, the measured values of $n_{Li}$ yield the following stoichiometric

compositions: $C_{42}LiC_{42}$, $C_{38}LiC_{38}$, $C_{18}LiC_{18}$, and $C_{14}LiC_{14}$ for stages I to IV, respectively (Fig. 3a), where the subscript in $C_N LiC_N$ corresponds to the number $N$ of carbon atoms in each of the two graphene layers per Li-ion (see Supplementary Methods 1.1 for details on how the compositions were assigned). Importantly, the inferred stoichiometries correspond to equidistant positions of Li ions at carbon hexagons' centers, which minimizes the Coulomb interaction energy (see Fig. 3b). Our experimental accuracy in determining $n_{Li}$ – which can be judged from the data scatter in Fig. 3a – is sufficient to unambiguously rule out stoichiometries any other than $N = 18$ and 14 for stages III and IV, respectively. Indeed, the nearest commensurable configurations ($N = 8$ and 24) correspond to greatly different $n_{Li}$. Even considering structures with broken hexagonal symmetry in Li-ion arrangements ($N = 12$, 16, and 20) (hence, unequal separations between Li ions and a larger interaction energy) would result in $n_{Li}$ well beyond our experimental error ($n_{Li} = 3.2$, 2.4, and $1.8 \times 10^{14}$, respectively). For less-doped stages I and II with the inferred $N = 42$ and 38, the nearest alternative commensurable configurations are $N = 32$ and 50, which would result in clearly different doping. However, in the latter case, we cannot fully exclude Li ion arrangements with broken hexagonal symmetries (e.g., $N = 36$).

To gain further insight into the observed in-plane staging, we used DFT calculations of the Gibbs free energy $\Delta G$ (relative to the unintercalated state) that can be written as (for details, see Supplementary Note 2.1)

$$\Delta G = \rho E_{int} - \rho \Delta\mu + k_B T \left[ \bar{\rho} \ln(\bar{\rho}) + (1-\bar{\rho})\ln(1-\bar{\rho}) \right], \quad (1)$$

where $E_{int}$ is the intercalation energy per Li-ion, $\rho = N_{Li}/N_C$ with $N_{Li}$ and $N_C$ being the number of Li and carbon atoms in the BLG supercell, and $\bar{\rho} = \rho N_C / N_{sites}$ (here $N_{sites}$ is the density of lattice sites available for Li ion intercalation). Equation 1 includes not only changes in the internal energy during intercalation of BLG but also the configurational entropy for ion insertion, where the third term takes into account the difference in the available intercalation sites, $N_{sites}$, for AB and AA stacking. This term becomes important at low doping levels (Supplementary Note 2.1 and Supplementary Fig. 8a). The second term in Eq. 1 arises because of the difference $\Delta\mu$ in chemical potentials of Li ions in the source (electrolyte) and within

the intercalated BLG. The value of $\Delta\mu$ can be determined experimentally as the drop in the pseudo-reference potential $V_{ref}$ (Fig. 1a) from the start of Li ions' entry into the bilayer to full intercalation. We found $\Delta\mu = 0.4 \pm 0.02$ eV (Supplementary Note 2.1 and Supplementary Fig. 7a). The contribution from the second term increases proportionally to $n_{Li}$ as per Eq. 1 and is most important for stages III and IV (Supplementary Fig. 8b–d).

The results of our calculations for $\Delta G$ are summarized in Fig. 6 where we focus on Li-ion configurations with high symmetry and, especially, the hexagonal ones that provide equally spaced Li ions and, therefore, lowest contributions to the Coulomb energy (Supplementary Note 2.1). Equidistant Li configurations, shown as bright symbols in Fig. 6, provide local minima in $\Delta G$ with respect to a multitude of other possible more disordered configurations, even if Li ions are allowed to reside only in carbon hexagons' centers ($\Delta G$ for some of the latter configurations are shown as semi-transparent symbols). Furthermore, Fig. 6 shows that the global minimum in the Gibbs energy, which represents the thermodynamic equilibrium for intercalated BLG, occurs at $N = 14$ ($n_{Li} \approx 2.7 \times 10^{14}$ cm$^{-2}$), in excellent agreement with the experimentally observed stoichiometry for in-plane stage IV. Note that this Li-ion density (storage capacity) is considerably (2.4 times) lower than that reached for stage-1 intercalation of bulk graphite[18–20]. An equally densely packed configuration for BLG, $C_6LiC_6$, is energetically unfavorable, presenting a very significant energy loss. Qualitatively, the lower saturated Li density achievable in BLG can be understood as the result of weak screening of interionic Coulomb repulsion[43] by the two graphene sheets, which makes it energetically costly for Li ions in BLG to reside as close to each other as in graphite. For confirmation, we calculated the intercalation energy for stage-1 graphite and $C_6LiC_6$ composition in BLG (Supplementary Note 2.1). This yielded a considerable difference of ~ 20 meV per Li-ion between intercalation energies for bulk graphite and BLG, respectively. A more extended discussion of the effect of screening can be found in Supplementary Note 2.4.

According to the energy diagram in Fig. 6, AB stacking remains energetically favorable only at low doping ($n_{Li} < 0.7 \times 10^{14}$ cm$^{-2}$) whereas the broad energy minimum for this stacking occurs at somewhat higher Li densities, between ~ 0.7 and $0.9 \times 10^{14}$ cm$^{-2}$. Although AB stacking seems energetically unfavorable with respect to

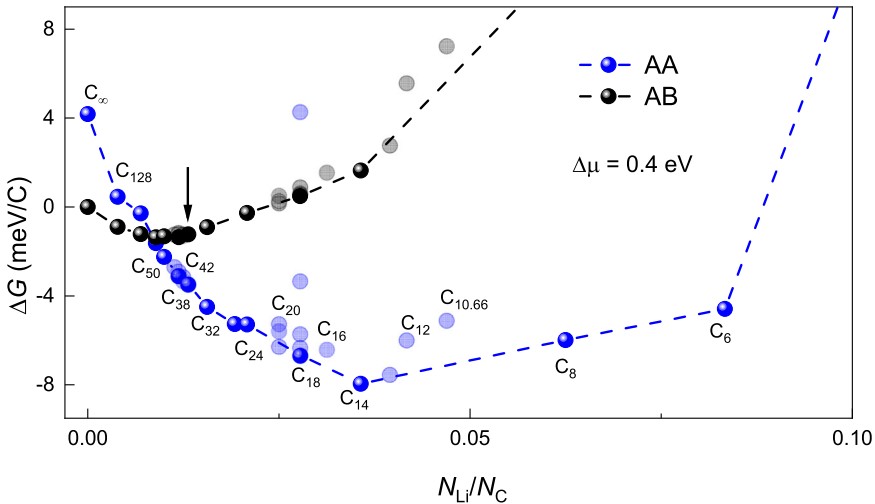

**Fig. 6 | Calculated free energy for AB- and AA- stacked bilayer graphene intercalated with lithium.** Data for AB- and AA- stacking are shown by black and blue symbols, respectively. Temperature 330 K and $\Delta\mu = 0.4$ eV, as determined experimentally. Labels mark $N$ in the corresponding $C_N LiC_N$ stoichiometries. Bright symbols correspond to Li-ion configurations with hexagonal symmetry; light symbols to non-hexagonal configurations with non-equidistant Li positions such as,

e.g., $\sqrt{21} \times 4$ for $C_{40}LiC_{40}$, $\sqrt{13} \times \sqrt{7}$ for $C_{20}LiC_{20}$. Where several possible configurations were considered for the same stoichiometry (e.g., $C_{42}LiC_{42}$, $C_{20}LiC_{20}$, $C_{18}LiC_{18}$) the hexagonal ion arrangement was always found to have the lowest energy. Dashed lines: guides to the eye connecting data for the hexagonal arrangements. The arrow indicates $N_{Li}/N_C$ at which the transition from AB to AA stacking is calculated to occur. Source data are provided as a Source Data file.

AA stacking for doping levels where the in-plane staging was observed ($n_{Li} \geq 0.9 \times 10^{14}\,cm^{-2}$), note that no gradual transition between AB and AA is expected during intercalation because the transition requires nucleation of AA domains within the AB bilayer, which involves local strain. To create such an AA domain in AB-stacked BLG, the gain in intercalation energy $\Delta E(r) = -\sigma A(r)$ should exceed the energy penalty associated with elastic deformations along the domain perimeter, $\gamma C(r)$, where $r$ is the radius of a domain, and $A$ and $C$ are its area and circumference, respectively. The DFT results yield $\sigma(n_{Li}) \approx 2.6\,meV/Å^2$, and $\gamma$ was estimated as $\sim 0.16\,eV/Å$ (Supplementary Note 2.2). This corresponds to the critical domain size $r_c = 2\gamma/\sigma \approx 12\,nm$, see Supplementary Fig. 9. The energy barrier for creating such critical-size precursors should postpone the transition from AB into AA stacking and allow the observation of metastable states at $n_{Li} > 0.7 \times 10^{14}\,cm^{-2}$ until Li density reaches the value $\sim 1.0 \times 10^{14}\,cm^{-2}$ as indicated by the arrow in Fig. 6. Therefore, we attribute in-plane stages I and II to the broad energy minimum seen in Fig. 6 and Supplementary Fig. 8b–d for the AB stacking (black curves). This corresponds to stoichiometric doping with $N = 40 \pm 2$ and agrees well with the experimentally found $N$ (Fig. 3a). The above results also allow us to understand the step-like changes in Li concentration from $\sim 10^{14}$ to $>2 \times 10^{14}\,cm^{-2}$ as the structural transition from AB to AA stacking, which is also accompanied by relaxation of Li ions into energetically preferable configurations commensurate with the underlying graphene lattice. In Fig. 6, this transition corresponds to the move between the main minima on the black and blue curves.

The origin of the staging around these two main equilibria (that is, the differences between stages I and II and stages III and IV) is less clear. Our DFT calculations in Fig. 6 yield that stages I and II (attributed to commensurate Li configurations with $N = 42$ and $38$, respectively) have close energies and, therefore, it is reasonable to expect that these energy minima can be occupied depending on fine details of the intercalation process. Because stage I is observed mostly during early intercalation cycles whereas stage II occurs after multiple cycles (Fig. 3a), we speculate that the elastic strain induced during intercalation-deintercalation leads to a slight shift between the two local energy minima around the global one for AB stacking. The strain appears because of the inevitable formation of AB and BA domains during restacking of the graphene layers from AA configuration back into AB configuration during deintercalation (as previously visualized for the case of thermal cycling of BLG[46]), see schematic in the inset of Fig. 4. Furthermore, AB/BA boundaries and, particularly, their intersections containing AA-stacked areas[46–48] can serve as nuclei for restacking AB bilayers into AA configuration, which are needed for reaching intercalation stages III and IV. The development of the extensive network of AB/BA boundaries can also contribute to the 'initial training' effect observed in our experiments in the first few intercalation cycles as well as an analogous behavior known for graphite-based electrodes[49,50]. Indeed, AB/BA boundaries exhibit a larger spacing between the two graphene layers[16], which reduces interlayer adhesion and the energy barrier for Li entry and diffusion[21,25], thus allowing the system to quicker reach stable configurations in subsequent cycles. This can explain why, after several cycles, the time spent during intercalation in AB stacking becomes shorter (Fig. 2c). It is also instructive to note that AB/BA boundaries contain 1D electronic channels with a finite carrier density even at the graphene's neutrality point[47]. Accordingly, this metallic network developed after multiple cycles should provide a notable electrical conductance within the poorly conducting deintercalated state. This additional conductance explains the counterintuitive observation discussed above that $\rho_{xx}$ in the fully deintercalated state decreased after each cycle.

As for stages III and IV that occur within the AA stacking, the DFT calculations yield that $N = 18$ and $14$ are also rather close in energy (Fig. 6). Because stage III was often found gradually transforming into stage IV (Supplementary Fig. 3c), we suggest that stage III ($N = 18$)

corresponds to a long-lived metastable state. The slow transition between stages III and IV also agrees with the fact that Li ions experience radically different diffusion barriers for AB and AA stacking, which are calculated as $\sim 70$ and $280\,meV$, respectively (Supplementary Note 2.3 and Supplementary Fig. 10). These values suggest rapid diffusion and, hence, sharp transitions between stages I and II whereas exponentially longer times can be required to reach local equilibria in the case of AA stacking (that is, to move between stages III and IV).

We emphasize that an important difference between the intercalation of graphite and of our small, defect-free graphene bilayer devices is the timescale. In graphite, intercalation typically takes hours, while in BLG Li ions fill the whole device in one step due to ultrafast diffusion[5]. In principle, one could expect the bilayer to go through all the configurations identified in our DFT calculations as energy minima (Fig. 6), but we only see the most stable ones, with 'jumps' between them due to significant Li density differences and ultrafast diffusion. The 'jumps' happen at a constant gate voltage due to finite energy barriers between different $C_xLiC_x$ configurations: the largest barrier is for AA domain formation and restacking from AB to AA and an appreciable barrier for Li diffusion through the AA-stacked bilayer. The time span of each $\rho_{xx}$ plateau then depends on the value of overpotential (Supplementary Fig. 4) and, with repeated cycling, is also affected by developing non-uniformities as discussed above. Abrupt jumps only occurred between stage I/II (AB stacked bilayer) and stage III/IV (AA stacked), supporting this explanation. The transition from stage III to stage IV was typically smooth (Supplementary Fig. 3c), while a transition from stage I to stage II for the same device was observed only once (Fig. 3a).

Our work was initially motivated by the lack of understanding of what determines the limits for Li intercalation in bilayer graphene. Although the answer may seem disappointing for potential applications, it is important that future developments take into account that the superior conductivity, large surface area, and ultrafast Li diffusion in potential ultrathin graphene electrodes would be tempered by a reduced Li storage capacity. This is particularly relevant for dense assemblies of BLG considered for battery technologies, which could provide a larger storage capacity than the one observed for isolated bilayers.

On a more fundamental level, we have identified previously unknown essential characteristics of the intercalation process. We have demonstrated that intercalation occurs in AB-stacked bilayers without immediate restacking to the AA configuration; AA restacking requires achieving a finite, rather large, Li-ion density and is itself required to achieve saturation in Li content. The two stages for each stacking order (AA and AB) involve relatively small changes in Li concentrations and are attributed to local equilibria that are close in energy and occur either as metastable states or because of shifting equilibrium conditions during intercalation cycles. We find that BLG can provide only weaker screening of interionic interactions compared to bulk graphite, so Li ions interact strongly and start repelling each other at longer distances, limiting the storage capacity of BLG. Another surprising finding is the experimental evidence for highly ordered Li configurations (essentially Li ion superlattices) which is of interest for electronic transport properties. It would be interesting to visualize the suggested $C_xLiC_x$ configurations by other techniques, especially scanning tunneling microscopy.

## Methods

### Device fabrication

To fabricate BLG devices, such as shown in Fig. 1b, a bilayer graphene crystal was mechanically exfoliated from bulk graphite and transferred onto a Si/SiO$_2$ (290 nm) substrate. This was followed by the

deposition of metal contacts using standard electron-beam lithography with poly(methyl methacrylate) (PMMA) resist and e-beam evaporation of Cr (3 nm)/Au (40 nm). In the next step a similar lift-off procedure was used to fabricate a large counter electrode made of Ta (3 nm)/Pt (40 nm) at ~250 μm distance from the BLG and two pseudo-reference electrodes, also Pt, see Supplementary Fig. 1a. Finally, the bilayer was shaped into a Hall bar geometry using a PMMA etch mask and reactive ion etching in oxygen plasma. To ensure that only a small part of the bilayer is exposed to the electrolyte during intercalation, we used a protective layer of SU-8 3005 (~5 μm thick) as shown in Fig. 1b and Supplementary Fig. 1a. Owing to the excellent chemical stability and insulation properties of SU-8[51], the device and the Cr/Au contacts were protected from unwanted electrochemical reactions that could take place due to contact with the electrolyte. The design of the SU-8 layer also ensured that Li ions from the electrolyte could enter the bilayer only through the exposed edge of the device.

### Electrolyte preparation and characterization

We used solid polymer-based Li-ion electrolyte PEO-LiTFSI[52,53]. The electrolyte was prepared in an Ar-filled glovebox. Prior to mixing the ingredients, a lithium salt, bis(trifluoromethanesulfonyl)imide (LiTFSI), and poly(ethylene oxide) (PEO) ($M_W = 100,000$ g/mol) were dried overnight at 170 °C and 55 °C, respectively, and acetonitrile was dehydrated with 3 Å molecular sieves at room temperature. Then 0.10 g LiTFSI and 0.31 g PEO were mixed with 2 mL acetonitrile under continuous stirring at room temperature for at least 24 h, giving the molar ratio of ethylene oxide to lithium (EO/Li) of 20:1 and a viscosity suitable for drop-casting. To this end, 2 μL of the pre-mixed electrolyte was drop-cast over the device using a micropipette, ensuring that the electrolyte covered the area encompassing the Pt counter and pseudo-reference electrodes, the exposed part of the bilayer graphene and a part of the SU-8 protective layer, but none of the exposed Au contact pads. The electrolyte was solidified through evaporation of acetonitrile to form Li ion conductive solid polymer electrolyte. The ionic conductivity of this electrolyte is due to Li ions moving under the applied voltage as they migrate between oxygen cites on the PEO backbone[53]. The prepared electrolyte was characterized using standard impedance spectroscopy[53] in a Pt–electrolyte–Pt two-probe configuration in a frequency range of 10 Hz – 500 kHz. Typical Nyquist plots of the electrolyte impedance at several different temperatures are shown in Supplementary Fig. 1c, d. In addition, we used impedance spectroscopy to determine the internal resistance of our devices, which was found to be $R_0 \approx 20$ kΩ (see Supplementary Fig. 1b for an example). To analyze the measured spectra, the electrochemical system was modeled as an equivalent electrical circuit shown in Supplementary Fig. 1b, using ZView® software. Measurements at temperatures between 30 °C and 60 °C showed that the ionic conductivity of our electrolyte is strongly temperature-dependent, increasing 50 times as temperature increased from 30 °C and 56 °C. Increasing the temperature further, from 56 °C to 60 °C, had a much smaller effect. Taking into account the melting point of the electrolyte, $T_m = 65$ °C, we have chosen 57 °C (330 K) as optimal for our intercalation/deintercalation measurements.

## Data availability

The authors declare that the data supporting the findings of this study are available within the paper and its Supplementary Information/Source Data file. Source data are provided in this paper.

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

## Acknowledgements

We acknowledge financial support from Horizon 2020 Graphene Flagship Project (Core 2 and Core 3), the Lloyd's Register Foundation, and the Engineering and Physical Sciences Research Council (EPSRC; Grant Nos EP/V007033/1 and EP/Z531121/1). T.A. and M.H. acknowledge support from the EPSRC CDT in Science and Applications of Graphene and Related Nanomaterials (EPSRC Grant EP/L01548X/1). We are grateful to Prof. Robert Dryfe for helpful discussions.

## Author contributions

I.V.G. and V.F. conceived and supervised the project. R.Z. and T.A. set up and carried out magneto-transport measurements; T.A., R.Z., and I.V.G. analyzed the results; D.D. designed and implemented Raman spectroscopy setup, supervised electrochemical and Raman measurements and contributed to the analysis of the results; Q.G., M.H., A.S., R.Z., Z.W., and K.I. fabricated the devices; J.M.H. performed DFT calculations and analysis; Z.A.H.G. and S.S. contributed to DFT calculations and analysis; V.F. supervised DFT calculations and theoretical analysis; I.V.G., A.K.G., J.M.H., and V.F. wrote the paper, with contributions from T.A. and R.Z. All authors contributed to discussions of the results.

## Competing interests

The authors declare no competing interests.
