## [Transparent Peer Review file · Nature Communications]

In-plane staging in lithium-ion intercalation of bilayer graphene

Corresponding Author: Professor Irina Grigorieva

Version 0:

Reviewer comments:

Reviewer #1

(Remarks to the Author)

The author introduces a compelling in-plane staging in BLG and conducts meticulous electrical measurements based on the Hall resistance of the fabricated device. However, there are concerns about the accuracy of determining lithium concentration solely through carrier density measurement. Additionally, the absence of direct observation of in-plane AA/ABAA-stacked areas, though beyond the paper's scope, is noteworthy. Before considering further publication, the following questions need addressing:

- 1) What are the device-to-device variances in lithium concentration determined by the Hall effect? The accuracy of carrier density and lithium concentration relies heavily on the equation $n_{Li} = (n + |n_0|)/0.9$, often affected by in-plane defects in few-layer graphene.
- 2) The authors mention testing five devices; please provide V_{xx} and V_{xy} data measured in all other devices in the supplementary information (SI). This is crucial for explaining device-to-device variances.
- 3) What is the reference electrode used in this device? It is recommended to check the potential difference between the reference electrode and BLG during intercalation/de-intercalation. Using V_{ref} to determine lithium concentration could be more accurate, especially in non-equilibrium conditions. Directly estimating the amount of lithium intercalated/de-intercalated into BLG can be achieved by monitoring cathodic/anodic current (between BLG and counter electrode) of the device.
- 4) The Nyquist plots of the solid electrolyte in Figure S1 show a substantial impedance value in the Pt–electrolyte–Pt two-probe configuration. The author should measure the impedance between BLG and counter/reference electrode to determine the internal resistance of the device.
- 5) The solid electrolyte will move back and forth between BLG and the counter under the applied gate voltage. The author should mention how this movements of electrolyte can affect the reversibility of lithiation and de-lithiation measurements?
- 6) The electrochemical stable window of the electrolyte (LiTFSI in PEO) is not wide enough to resist decomposition between -7 V to 0 V. The authors should discuss the effect of electrolyte decomposition and SEI formation on the reversibility of the devices.

Reviewer #2

(Remarks to the Author)

This manuscript by Astles and co-workers describes in-operando intercalation dynamics of Li ions in bilayer graphene. The authors manage to observe four different intercalation stages in the process, and suggest the hypothesis that an AB to AA stacking transition takes place during the Li intercalation, which is an original, interesting claim.

The work is original and well prepared, with a good combination of experimental and modelling work, and shall be published after some revision.

The key parameter of the work, i.e. Li ion density in the bilayer, is estimated by measuring charge density electrically, measuring graphene resistivity and Hall voltage. Charge density calculations have to rely on some assumptions; as example, the initial p-doping of graphene, which is in the best case ca 10% of the total signal to be measured, should be removed. Also, assumptions about constant scattering time for stages II and III need to be made. While the measurements are interesting, having an independent way to measure this parameter would strengthen much more the conclusions. Raman measurements have been performed on similar systems to measure Li and Na intercalation in operando; see as example Nano Letters 2018 18 (1), 460-466 and Sci. Adv. 2021; 7 : eabf0812. By correlating G band shift and 2D band intensity, additional insight on the doping could be obtained.

Domain size of unperturbed graphene domains could be also estimated using Tuinstra-Koenig approximation, or even more complex formulas specific for point-like defects, and this could help to clarify the assumption of AA/AB domain wall formation, and the presence of strain around an AA-stacked domain.

I understand that the presence of SU8 coating on the graphene could be a problem for measurements, but other previous works have managed to overcome this problem, and I feel that Raman spectroscopy data could truly help to confirm the main conclusions. In alternative, the 2nd paper mentioned above also used imaging ellipsometry to monitor the intercalation of ions, but this is a much more specialized technique, which is maybe not available to the authors. Thus I suggest major revisions to use some independent technique to better confirm the main claims of the paper.

Minor revisions:

Fig. 4 very dense, difficult to connect the color coding to fig. 1c. add labels and move one of the insets to SI or to a different figure.

Page 7, clarify what is N_{sites} used in calculation of eq. 1

SI page 2 correct typo "p-doing".

Reviewer #3

(Remarks to the Author)

Manuscript number: NCOMMS-23-49125

Authors: Thomas Astles et al.

Title: In-plane staging in lithium-ion intercalation of bilayer graphene

In this work, the authors focus on the lithium intercalation mechanisms in bilayer graphene (BLG) using on-chip electrochemical cells. They manage to very precisely follow the (de)-lithiation of the their BLG electrode during several cycles by measuring operando the evolution of the graphene resistivity ρ_{xx} , as well as the density of lithium ions on given moment of the lithiation process. The interpretation of their measured value of lithium densities is supported with DFT computation of the Gibbs free energy of the BLG for different lithiation states and structures of the host lattice.

The paper is clearly written and organized. Detailed information regarding the device fabrication, electrolyte characterization as well as transport measurements interpretations and DFT computations are provided in the supplementary information.

The operando measurements are very interesting to the field. The DFT computations and the estimation of the nucleation of AA domain in initially AB-stacked BLG bring also interesting points to the understanding of this complex phenomenon. However, the source of lithium in the electrochemical set-up is unclear, which makes the understanding of the electrochemical behaviour of the set-up rather difficult to follow. Furthermore, the attribution of the 4 measured plateaus of the graphene resistivity directly to specific "stages" of the lithiated BLG is not sufficiently supported given that it is the main message of this work. For these reasons (and given some other comments below), and given the very high standard of the journal, I would recommend the publication of this work only after major revision.

Comments:

The operating principle of the galvanic cell is not detailed. It seems that the counter electrode (CE) is made only of platinum (Pt). In this case, what is the source for Li-ions for the first intercalation? In the electrochemical cell, what is the reference potential for the gate voltage V_g ? The chemical potential of Li in C_6LiC_6 is around 0.15V versus Li/Li⁺, and is around 1.5V for the delithiated state of the BLG [1]. How do you explain the characteristic voltage gap potential of roughly -3V where the intercalation start?

The graphene resistivity, ρ_{xx} , is followed during all the intercalation/de-intercalation. Four different plateaus are observed on the measurements (after some "training cycles"), and this on a rather reproducible way. The density of the li-ions, n_{Li} , is obtained from the carrier density in the BLG n , which is obtained from the Hall voltage. It is measured only at certain moment during the intercalation, especially on the different plateaus.

The four plateaus of the graphene resistivity, ρ_{xx} , are attributed to four distinct states of lithiated BLG, referred to by the authors as in-plane stages I to IV. In lithiated graphite, during intercalation, plateaus are observed on the cell potential. They are not attributed to one phase, but on the contrary, they are the sign of the co-existence of two distinct phases. During a plateau, the chemical potential of the two phases remains constant and it is the ratio of the two phases that evolves. What are the arguments to interpret the plateau in ρ_{xx} as the sign of only one in-plane stage? In this latter case what would be the dynamic of transition between the different stages during the intercalation process? Could you measure the evolution of the density of the li-ion, n_{Li} , during a whole lithiation step, to monitor the lithium content of the whole BLG electrode? It seems surprising that the state of lithiation of the BLG electrode "jumps" from one state of lithiation to the next one, without period of phase co-existences, given the constant applied voltage gap potential and the different lithium content of the

different “stages”.

The determination of the most favorable lithium configuration from the Gibbs free-energy is usually determined with the common tangent construction (or convex hull) between the different energy minima, leading to phase separating regions and coexistence of different stages [3]. With this in mind could the author explain why the system does not go to the configuration AA C32, which seems well on the convex hull of the computed Gibbs free-energy and would directly go to AA C18 or 14 ? Staging usually refers to the formation of staged structure, which are defined as “periodic sequences of host and guest layers, with n layers of the host material separating neighboring guest layers in a stage n intercalation compound. In some cases, the staged structures are three-dimensionally ordered crystals, but very often this is not the case. Thus staging is commonly regarded as being a quasi-one-dimensional ordering phenomenon.” [REF 2]. The fact that the authors refer to lithium ordering in the plane between the two adjacent graphene layers as “intercalation stages” is somehow confusing, as there is no “staging” phenomena, but different lithium ordering.

[1] Kühne et al., Nature nanotech. 12 895-900, 2017

[2] Kirczenow, G. (1990). Staging and Kinetics. In: Zabel, H., Solin, S. (eds) Graphite Intercalation Compounds I. Springer Series in Materials Science, vol 14. Springer, Berlin, Heidelberg.

[3] Smith et al., J. Phys. Chem. C 2017, 121, 12505–12523

Reviewer #4

(Remarks to the Author)

The current manuscript reports on the mechanism of lithium (Li) ions intercalation in bilayer graphene (BLG) systems. The work shows that the intercalation on BLG occurs via four distinct stages (in-plane stages I to IV), which is different from what occurs on graphite. The work also shows that the maximum Li ion density that can be inserted in BLG systems should be at the order of 2.7×10^{14} , which can be achieved at stage IV and correspond to the stoichiometric compound C14LiC14.

However, an essential step to achieve the highest Li storage for BLG is attributed to AB to AA stacking transition, which is facilitated by the AB/BA boundaries formed during intercalation-deintercalation cycles.

The work was conducted very carefully and the reproducibility of the data is impressive, being observed in many different fabricated devices. The agreement between experiment and the DFT calculations in this work is also very relevant and assisted in providing robustness to the experimental data and the conclusions taken in the work. Hence, I do strongly recommend the paper for publication. I suggest the following questions points for further elaboration and additional discussion :

1) References 7 and 8 show that the denser arrangements should be expected for Li ions intercalation on BLG, corresponding to the compound C6LiC6 in contrast to the densest arrangement found in stage IV of this work which correspond to the stoichiometric composition of C14LiC14. Reference 7 shows compelling evidence via in-situ TEM that even close-packed metallic Li could be intercalated in between the layers and that the conclusions presented in the paper should hold irrespective to the stacking order. Reference 8, in turn, shows that there is no fundamental difference between bilayer or few-layer graphene with respect to Li storage manner and kinetic behavior and that the saturated composition should be C6LiC6. The authors should, therefore, clarify these apparent contradictions. Why the configuration that would give the highest Li density is C14LiC14 instead of C6LiC6, such as in the previous reports?

2) Since previous reports have shown that Li ion density achievable in BLG systems is about 3 times lower than the one achieved in bulk graphite and considering that it is hard for BLG to replace graphite in practical applications, the authors should clarify on the motivation and significance of the work.

3) In the paragraph contained in lines 196 to 209, the authors should explain how they assigned the stoichiometric compositions of C42LiC42, C38LiC38, C18LiC18 and C14LiC14 to stages I to IV respectively. The authors should consider, perhaps in the supplementary material, to show in a more careful how to assign the number N of carbon atoms to the Li ion densities measured. Moreover, it would be highly valuable and it would make the results more robust if the authors could provide experimental validation of the stoichiometries of the lithiated products, such as C14LiC14, via Raman, XPS, EELS or any other experimental data or at least comment on feasibility .

4) The manuscript highlights the difference in lithium ion behavior between bilayer graphene (BLG) and bulk graphite, attributing it to the weaker screening effect in thin layers. However, in reference 35, which was one of the references cited in the work to back up this argument, the authors even with the reduced screening potential, the most stable Li density should be approximately LiC7, which is different from the LiC14 found in this work. The authors should comment on this .

Moreover, the screening argument raises a question about the transition from 2D to bulk behaviour: At what point, in terms of the number of graphene layers, does this 2D screening effect diminish sufficiently for lithium ions to behave similarly to those in bulk graphite, without significant repulsion? Clarifying this aspect would provide valuable insight into how the properties of graphene-based materials evolve from the nanoscale to the bulk scale. In reference 34, it seems that the screening effect changes significantly from the monolayer to bilayer and to the rest (few-layer and bulk), but the screening in the bilayer does not significantly changes compared to few-layered or bulk systems?

5) Finally, the authors should comment why the Li ions are expected to predominantly intercalate within the interlayer of BLG and why surface placement on the bilayer can be neglected. ?

Author Rebuttal letter:

REPLY TO REVIEWERS

Reply to comments of Reviewer #1:

The author introduces a compelling in-plane staging in BLG and conducts meticulous electrical measurements based on the Hall resistance of the fabricated device. However, there are concerns about the accuracy of determining lithium concentration solely through carrier density measurement. Additionally, the absence of direct observation of in-plane AA/ABAA-stacked areas, though beyond the paper's scope, is noteworthy.

We thank the Reviewer for the insightful comments and suggestions. Our replies are below:

Before considering further publication, the following questions need addressing:

1) What are the device-to-device variances in lithium concentration determined by the Hall effect? The accuracy of carrier density and lithium concentration relies heavily on the equation $\delta Li = (\delta + |\delta_0|)/0.9$, often affected by in-plane defects in few-layer graphene.

Indeed we rely on the value of charge transfer per Li ion (0.9e⁻) which could not be independently measured but was consistently found in several DFT studies in literature (e.g. refs. [11, 15] in the revised manuscript) and in our own DFT calculations. In literature the DFT charge transfer value ranges from 0.85 to 0.88 to 0.9e⁻; we are using a rounded value of 0.9e⁻ throughout. Using a lower value of 0.88e⁻ would result in just 1-2% difference in nLi, which is below our experimental error and will not affect the inferred Li ion configurations or other conclusions. Consistency in Li ion concentration nLi calculated from measurements of carrier density on different devices is illustrated in Fig. 3a; the fact that these nLi show good agreement with discussed Li ion stoichiometries is a further confirmation that they are extracted correctly. We emphasise this in the revised manuscript.

With regard to a possible effect of in-plane defects, we considered different types of defects: point defects, AB/BA boundaries. We did not find any evidence of point defects in Raman spectra (see our reply to reviewer #2): there was no D peak above the noise level either before, during or after intercalation. It is still possible that a very small D peak (below the noise level) is present, so we estimated the upper limit on the defect density from the $\delta^{1/4}D/\delta^{1/4}G$ ratio. In deintercalated state, assuming $\delta^{1/4}D/\delta^{1/4}G \approx 30$ (noise level), we obtain $\delta^{1/4}D/\delta^{1/4}G < 0.02$ and the upper limit on the defect density $\delta D < 7.5 \text{ \AA} \times 10^9 \text{ \AA}^{-2} \approx 7.5 \times 10^9 \text{ cm}^{-2}$, insignificant compared to our nLi (here EL=2.4 eV is the laser excitation energy [Cancado et al Nano Lett. 11, 3190-3196 (2011)]). As concerns AB/BA boundaries, they clearly do not serve as sinks for charge because $\tilde{\rho}_{xx}$ and nLi are reproduced in different cycles and no significant (more than a few percent) change in n0 was found after 15 cycles of intercalation. If these boundaries did affect nLi, we would not expect such reproducibility. This information has been added in the revised manuscript.

2) The authors mention testing five devices; please provide Vxx and Vxy data measured in all other devices in the supplementary information (SI). This is crucial for explaining device-to-device variances.

Following the Reviewer's suggestion, we have included a new plot in the revised SI (new Figure S6). To compare different devices, we show the evolution of $\tilde{\rho}_{xx}$ in the 5 devices for representative (middle) cycles, where staging has already been established. Using the $\tilde{\rho}_{xx}$ rather than Vxx allows us to account for different dimensions of the devices (widths, distances between the contacts). The qualitative evolution of cycling is reproduced on all studied devices although for some of the devices the intermediate plateaus (stages II and/or III) were missing in all studied cycles. The reasons for the absence of some of the stages are discussed where we compare our experimental data with theory; we have further clarified this point in the revised manuscript. Additionally, the time duration of the more dilute stages was sometimes very short (e.g., device E), where the bilayer transitioned very quickly from stage I to stage IV. In our later experiments we found that one of the reasons for this is the value of overpotential, i.e., by how much the applied Vg exceeded the threshold value for the device (Fig. S4).

1

3) What is the reference electrode used in this device? It is recommended to check the potential difference between the reference electrode and BLG during intercalation/de-intercalation. Using Vref to determine lithium concentration could be more accurate, especially in non-equilibrium conditions. Directly estimating the amount of lithium intercalated/de-intercalated into BLG can be achieved by monitoring cathodic/anodic current (between BLG and counter electrode) of the device.

We thank the Reviewer for this suggestion. The reference electrodes for our devices are Pt, same as the gate electrodes, so it is more appropriate to call them pseudo-reference; we have clarified this in the revised manuscript. Following the Reviewer's suggestion, we have estimated the amount of Li entering the device from a step in the cathodic current between the counter electrode and BLG (new Figure S7b). This current step corresponds to a step in $\tilde{\rho}_{xx}$. Integrating the current over the step gives a charge $\Delta Q \approx 4 \times 10^{-10} \text{ C}$, i.e., entry of $\Delta n \approx 5 \times 10^{14} \text{ cm}^{-2}$ electrons, in good order-of-magnitude agreement with the number of intercalated Li ions corresponding to the transition from stage I to stage III in this case ($\sim 1.3 \times 10^{14} \text{ cm}^{-2}$, see Fig. 3a). Here $e = 1.6 \times 10^{-19} \text{ C}$ is the electron charge and $\Delta A \approx 5 \times 10^{-6} \text{ cm}^2$ the area of our $\sim 10 \mu\text{m} \times$

50 Åm device. We have added this estimate and a brief discussion in the revised manuscript. We have also attempted to estimate the amount of intercalated Li from the δ_{ref} at 0.4 V in Fig. S7a. This however requires a reliable estimate of the capacitance of the device which depends on several unknown factors.

4) The Nyquist plots of the solid electrolyte in Figure S1 show a substantial impedance value in the Pt electrolyte-Pt two-probe configuration. The author should measure the impedance between BLG and counter/reference electrode to determine the internal resistance of the device.

As requested, we have measured the impedance between the counter electrode and the BLG, see new Fig. S1e. The internal resistance of the device is indeed high, δ_0 at 20 k Ω . We note, however, that due to very small anodic/cathodic currents in our experiments $\delta_{1/4} < 10$ nA, this corresponds to a voltage drop of < 0.2 mV, negligible compared to the voltage drop due to the applied gate voltage. We have clarified this point in the revised manuscript.

5) The solid electrolyte will move back and forth between BLG and the counter under the applied gate voltage. The author should mention how this movements of electrolyte can affect the reversibility of lithiation and de-lithiation measurements?

We did not see any visible evidence of the electrolyte movement when inspecting the devices before and after intercalation. If there was such movement, it would probably damage the contacts, which did not happen. This agrees with the general understanding of ion conductivity of this particular electrolyte where only the Li ions move under the applied voltage as they migrate between oxygen sites on the PEO backbone (see e.g. Vahva et al, ChemElectrochem 2021, 8, 1930-1947). We have mentioned this in the revised Supplementary Information.

6) The electrochemical stable window of the electrolyte (LiTFSI in PEO) is not wide enough to resist decomposition between -7 V to 0 V. The authors should discuss the effect of electrolyte decomposition and SEI formation on the reversibility of the devices.

We agree with the Reviewer that electrolyte degradation is an important issue. In our experiments we did not see significant changes in electrolyte conductivity/internal device resistance after 15-20 cycles, as confirmed by repeated impedance spectroscopy measurements. We believe that the reason for such stability is that a large fraction (30 to 40%) of the applied 7V potential difference falls at the Pt counter electrode, where no faradaic reactions are expected. The potential drop at the graphene interface (as found from our measurements of V_{ref} , potential difference between the pseudo-reference and graphene) is then < 4.5 V, comparable or less than the reported electrochemical stability window for LiTFSI in PEO, ~ 4.5 V vs Li/Li $^+$ (e.g., Mery et al, Materials 2021, 14, 3840). At least for the number of cycles used in our study (< 20) this relatively large gate voltage did not have an appreciable effect. As concerns SEI formation, we cannot rule it out, and it may have contributed to changes seen in the initial intercalation cycles (in addition to bilayer expansion and formation of AB/BA boundaries). Importantly, SEI formed at negative potentials is porous and not expected to prevent intercalation (Mery et al, Materials 2021, 14, 3840), in agreement with our observations. We have added a brief discussion of these effects in the revised manuscript.

2

Reply to comments of Reviewer #2

This manuscript by Astles and co-workers describes in-operando intercalation dynamics of Li ions in bilayer graphene.

The authors manage to observe four different intercalation stages in the process, and suggest the hypothesis that an AB to AA stacking transition takes place during the Li intercalation, which is an original, interesting claim.

The work is original and well prepared, with a good combination of experimental and modelling work, and shall be published after some revision.

We thank the Reviewer for highlighting the originality and careful preparation of our work.

The key parameter of the work, i.e. Li ion density in the bilayer, is estimated by measuring charge density electrically, measuring graphene resistivity and Hall voltage. Charge density calculations have to rely on some assumptions; as example, the initial p-doping of graphene, which is in the best case ca 10% of the total signal to be measured, should be removed. Also, assumptions about constant scattering time for stages II and III need to be made. While the measurements are interesting, having an independent way to measure this parameter would strengthen much more the conclusions. Raman measurements have been performed on similar systems to measure Li and Na intercalation in operando, see as example Nano Letters 2018 18 (1), 460-466 and Sci. Adv. 2021; 7 : eabf0812. By correlating G band shift and 2D band intensity, additional insight on the doping could be obtained.

Domain size of unperturbed graphene domains could be also estimated using Tuinstra-Koenig approximation, or even more complex formulas specific for point-like defects, and this could help to clarify the assumption of AA/AB domain wall formation, and the presence of strain around an AA-stacked domain. I understand that the presence of SU8 coating on the graphene could be a problem for measurements, but other previous works have managed to overcome this problem, and I feel that Raman spectroscopy data could truly help to confirm the main conclusions. In alternative, the 2nd paper mentioned above also used imaging ellipsometry to monitor the intercalation of ions, but this is a much more specialized technique, which is maybe not available to the authors.

Thus I suggest major revisions to use some independent technique to better confirm the main claims of the paper.

To address the Reviewer's request for an alternative way to monitor intercalation and estimate the amount

of intercalated Li, we have constructed a special custom cell and devised a new setup that allowed us to use Raman spectroscopy during intercalation and correlate the results with changes in device resistance. Unfortunately, this work delayed the resubmission by several months. The key challenge was the strong background signal from electrolyte (SU8 was somewhat less problematic). Once we were able to constrain the electrolyte to a very thin layer, we were able to collect meaningful spectra shown now in a new figure, Fig. 5 in the main text.

As seen in this figure, the G peak position shifted sharply by $\sim 8 \text{ cm}^{-1}$ as soon as the gate voltage exceeded the threshold value (3.8V for this device) while the longitudinal resistance (measured simultaneously) went through a sharp peak indicating intercalation. At the same time the 2D peak in the intercalated state was strongly suppressed. In addition, $I(2D)/I(G)$ intensity ratio changed from ~ 0.8 before intercalation to ~ 0.5 after. This behaviour indicates strong electron doping [A. Das et al, Nature Nanotech. 3, 210-215 (2008); A. Das et al. Phys. Rev. B 79, 155417 (2009), Sci. Adv. 2021; 7 : eabf0812] and further discussed below. Both peaks return to the initial position and intensity after deintercalation. The shown spectra were collected during the 2nd intercalation cycle and similar curves obtained for the 3rd cycle. Intercalation in this case reached in-plane stage I/II and carrier densities $n \sim 10^{13} \text{ cm}^{-2}$ (Fig. 3a). As expected, both G and 2D peaks remain visible for this doping, very similar to the case of stage-II Li intercalation of graphite where

doping was even higher (Inaba et al. J. Electrochem.Soc. 142, 20-26 (1995)). Although further cycling led to higher electron density, unfortunately, we could not quantify this evolution using Raman spectra because noise strongly increased, probably for the observed degradation of the electrolyte under the laser beam.

Nonetheless, the Raman data allowed new insight into the intercalation process. Firstly, we observed no D peak above the noise level either before, during or after intercalation. This indicates that Li ion intercalation did not induce point-like defects in graphene, in agreement with our conclusions from resistance measurements (no increase in resistance after deintercalation). We estimated the maximum possible defect density induced by intercalation assuming that the D peak was comparable in intensity with the noise level. This yielded $\delta D/G < 0.02$ which translates into minute possible densities of point-like defects of $< 5 \times 10^9 \text{ cm}^{-2}$ (see our reply to Reviewer #1).

Secondly, from the observed shift in the G peak position and the change in $I(2D)/I(G)$ intensity ratio we could place a lower limit on the electron doping induced by intercalation in our devices. As they were always significantly hole-doped in the unintercalated state ($n \sim 10^{13} \text{ cm}^{-2}$), the shift in the G peak position by $\sim 8 \text{ cm}^{-1}$ corresponds to electron doping much larger than 10^{13} cm^{-2} and larger than that observed by A. Das et al. [Phys. Rev. B 79, 155417 (2009)] for electrostatically gated bilayer graphene. The authors achieved $n \sim 5 \times 10^{13} \text{ cm}^{-2}$, the highest value reported in the literature for the experimental evolution of the G and 2D peaks in bilayer graphene but twice smaller than that for our in-plane stage I/II where the Raman spectra were taken. Unfortunately, the data scatter in Das et al and saturation in positions of the Raman peaks at high electron doping do not allow extrapolation to higher densities to estimate our doping levels.

Unfortunately, all we can say from this comparison is that we achieved a higher doping density than Das et al. The same conclusion comes from comparison with the G peak intensities in their paper.

It is also important to note that both G peak shifts and carrier densities extracted from the Hall resistance in our experiment agree well with those reported under Li ion intercalation of bilayer graphene [M. Kuhne's thesis, group of Jurgen Smet]. They found a G peak shift of 5-10 cm^{-1} and Li ion density from the Hall resistance of $\sim 10^{14} \text{ cm}^{-2}$, both values close to our findings.

As for using Raman spectra to detect AB/BA domain formation and/or restacking to AA, as suggested by the Reviewer, unfortunately AB/BA boundaries do not have a signature or cause a D peak in Raman spectra (see e.g. Barbosa et al, 2D Materials 9, 025007 (2022)). According to theory (Hertrich et al, arXiv:1806.06026), AB and AA stacked bilayers might be distinguishable from a subtle change in the 2D peak shape (we are not aware of any experimental work that would confirm this). Moreover, it would be difficult to detect such changes in our experiment. Indeed, we are dealing not with a perfect AA bilayer but rather with a highly-doped AA state that is also strongly strained because of domains and their boundaries. The main effect from this is expected to be a significant broadening and suppression of the 2D peak, as indeed observed in our experiment. This behavior is also similar to that observed for intercalation of graphite [Inaba et al]: although graphite is known to exhibit a change from AB to AA stacking during intercalation, no spectra showing this change were reported, despite having macroscopic samples for observation.

The new figure and discussion of the Raman data have been added to the revised manuscript (main text).

We are grateful to the Reviewer for suggesting Raman measurements. Although it took a lot of time and effort, the spectra have strengthened our conclusions and provided interesting additional info. The pointed-out paper Sci. Adv. 2021 has also been added to the manuscript.

Minor revisions:

Fig. 4 very dense, difficult to connect the color coding to fig. 1c. add labels and move one of the insets to SI or to a different figure.

We are sorry and grateful to the Reviewer for noticing this. Indeed the color-coding was inconsistent with Fig. 1c and the plot included deintercalation curves for the initial 3 cycles that are not shown in Fig. 1c. We have updated Fig. 4 to show data for the same cycles as Fig. 1c and the color coding is now consistent. All deintercalation curves are now shown in a new SI figure (Supplementary Fig. 5a).

Page 7, clarify what is N_{sites} used in calculation of eq. 1

N_{sites} is the density of lattice sites available for Li-ion intercalation. We have clarified in the revised

Corrected, thank you.

Reply to comments of Reviewer #3:

Manuscript number: NCOMMS-23-49125

Authors: Thomas Astles et al.

Title: In-plane staging in lithium-ion intercalation of bilayer graphene

In this work, the authors focus on the lithium intercalation mechanisms in bilayer graphene (BLG) using on-chip electrochemical cells. They manage to very precisely follow the (de)-lithiation of the their BLG electrode during several cycles by measuring operando the evolution of the graphene resistivity ρ_{xx} , as well as the density of lithium ions on given moment of the lithiation process. The interpretation of their measured value of lithium densities is supported with DFT computation of the Gibbs free energy of the BLG for different lithiation states and structures of the host lattice.

The paper is clearly written and organized. Detailed information regarding the device fabrication, electrolyte characterization as well as transport measurements interpretations and DFT computations are provided in the supplementary information.

The operando measurements are very interesting to the field. The DFT computations and the estimation of the nucleation of AA domain in initially AB-stacked BLG bring also interesting points to the understanding of this complex phenomenon.

We thank the Reviewer for this positive assessment of our work and thoughtful comments.

However, the source of lithium in the electrochemical set-up is unclear, which makes the understanding of the electro-chemical behaviour of the set-up rather difficult to follow. Furthermore, the attribution of the 4 measured plateaus of the graphene resistivity directly to specific "stages" of the lithiated BLG is not sufficiently supported given that it is the main message of this work. For these reasons (and given some other comments below), and given the very high standard of the journal, I would recommend the publication of this work only after major revision.

We accept the criticism. Please find our point-by-point replies below.

Comments:

The operating principle of the galvanic cell is not detailed. It seems that the counter electrode (CE) is made only of platinum (Pt). In this case, what is the source for Li-ions for the first intercalation? In the electrochemical cell, what is the reference potential for the gate voltage V_g ? The chemical potential of Li in C_6LiC_6 is around 0.15V versus Li/Li^+ , and is around 1.5V for the delithiated state of the BLG [1]. How do you explain the characteristic voltage gap potential of roughly -3V where the intercalation start?

We are sorry for not being sufficiently clear. The counter electrode we are using is indeed a Pt (ion-blocking) electrode. The source of Li ions is the LiTFSI salt in the electrolyte. As described in Supplementary Methods, we are using solid polymer electrolyte PEO-LiTFSI where Li ions move under the applied voltage as they migrate between oxygen sites on the PEO backbone (e.g., Vahva et al, ChemElectrochem 2021, 8, 1930-1947). The role of the counter electrode is to create a potential drop at the interface with BLG such that the chemical potential of Li in C_xLiC_x becomes lower than the chemical potential of "free" Li which in our case is in the form of mobile ions in PEO-LiTFSI electrolyte. The threshold potential for intercalation in this case does not correspond to a thermodynamic value (as in the case of Li/Li^+ electrode) and has to be determined experimentally for each device (as it depends on electrolyte preparation and other details). To compare the experimental results with our DFT calculations for the free energy, we defined a reference potential as $\delta_{ref} = \delta + \Delta\delta$ and calibrated δ by calculating the Gibbs free energy such that intercalation of Li ions is energetically disfavored for all Li densities, i.e., such that $\Delta\delta=0$ (corresponding Gibbs free energy as a function of Li density is shown in the inset of Supplementary Fig. 8b). In experiment $\delta_{ref} = \delta$ ($\Delta\delta = 0$)

5 corresponds to the start of intercalation as indicated by the sharp change in the device resistance. We found $\delta_{ref} \hat{=} 2.9$ V, that is $\delta_{ref} = \delta \hat{=} 2.9$ eV (Supplementary Fig. 7a). The applied δ_g in this case was -4.5V; the difference between δ_g and δ_{ref} is due to a significant voltage drop at the counter electrode (see our reply to Reviewer #1.) We have included this explanation in the revised Supplementary Information, which hopefully makes the paper clearer.

The graphene resistivity, ρ_{xx} , is followed during all the intercalation/de-intercalation. Four different plateaus are observed on the measurements (after some "training cycles"), and this on a rather reproducible way. The density of the li-ions, n_{Li} , is obtained from the carrier density in the BLG n , which is obtained from the Hall voltage. It is measured only at certain moment during the intercalation, especially on the different plateaus.

The four plateaus of the graphene resistivity, ρ_{xx} , are attributed to four distinct states of lithiated BLG, referred to by the authors as in-plane stages I to IV. In lithiated graphite, during intercalation, plateaus are observed on the cell potential. They are not attributed to one phase, but on the contrary, they are the sign of the co-existence of two distinct phases. During a plateau, the chemical potential of the two phases remains

constant and it is the ratio of the two phases that evolves. What are the arguments to interpret the plateau in ρ_{xx} as the sign of only one in-plane stage?

We measured the device resistance ρ_{xx} and ρ_{xy} at different pairs of contacts, i.e., in different sections of the device, and observe identical evolution of the resistance in different sections (Fig. 2a,b, Supplementary Fig. 2). If there were two phases present (unintercalated and intercalated ones, or domains with different Li ion densities as in graphite) we would observe uncorrelated steps at different contacts or, if domains were nanoscopic or the intercalation were disordered, a smooth evolution of ρ_{xx} (Van der Ven et al. Chem. Rev. 120, 6977 (2020)). Our observations are quite different: changes in Li density occurred simultaneously (with our time resolution) over the whole device. This rapid change is attributed to very fast Li diffusion facilitated by the fact that our devices were relatively small (typically $40 \times 5 \text{ \AA}^2$) and graphene bilayers were atomically flat and defect-free. Such fast diffusion agrees well with the earlier observations (e.g., KÅ¼hne et al. 2017 also mentioned by the Reviewer as ref. [1] below). Accordingly, the plateaus in ρ_{xx} and ρ_{xy} , which are highly reproducible for different cycles and different devices (see new Supplementary Fig. 6), should correspond to states with uniform Li density over the entire device, which we refer to as "in-plane stages". We have clarified this in the revised manuscript.

In this latter case what would be the dynamic of transition between the different stages during the intercalation process? Could you measure the evolution of the density of the Li-ion, n_{Li} , during a whole lithiation step, to monitor the lithium content of the whole BLG electrode? It seems surprising that the state of lithiation of the BLG electrode "jumps" from one state of lithiation to the next one, without period of phase co-existences, given the constant applied voltage gap potential and the different lithium content of the different "stages".

An important difference between graphite and our graphene bilayer is the timescale: in graphite full intercalation typically takes hours while in our small defect-free devices, Li ions fill the whole device in one step due to their fast diffusion, in agreement with the earlier studies (ref. [1] below). According to our DFT calculations, one could expect many more Li configurations (Fig. 6 in the revised manuscript) but, experimentally, only the four most stable ones showed up, basically because of small energy barriers between local energy minima for different C_xLiC_x configurations. The largest barrier corresponds to the change from AB to AA configurations. Furthermore, the time span of each ρ_{xx} plateau depended on the value of overpotential (as demonstrated in Supplementary Fig. 4) and, with repeated cycling, also started to be affected by developing non-uniformities, such as residual Li ions present in the bilayer after deintercalation. This explanation is supported by our observations that the abrupt jumps/steps only occurred between stage I and stage III, OR stage II and stage III, OR stage I and stage IV - that is, between one of the stages with AB stacking and one with AA stacking. Therefore, the Reviewer is right: we do see coexistent phases in some of the measurements. For example, the intercalation transition from stage III to stage IV was typically smooth (Supplementary Fig. 3c) and, also, deintercalation often occurred smoothly (inset of Fig. 4). Moreover, we saw switching back and forth between AA and AB configurations that occurred either smoothly or through some poorly defined intermediate states (Fig. 1e, Supplementary Fig.

6

3a). All these occurrences indicate a mixture of different local configurations. Admittedly, this point was explained insufficiently in the original manuscript and we have done our best to improve the presentation. The determination of the most favorable lithium configuration from the Gibbs free-energy is usually determined with the common tangent construction (or convex hull) between the different energy minima, leading to phase separating regions and coexistence of different stages [3]. With this in mind could the author explain why the system does not go to the configuration AA C32, which seems well on the convex hull of the computed Gibbs free-energy and would directly go to AA C18 or 14 ?

[1] KÅ¼hne et al., Nature nanotech. 12 895-900, 2017

[2] Kirczenow, G. (1990). Staging and Kinetics. In: Zabel, H., Solin, S. (eds) Graphite Intercalation Compounds I. Springer Series in Materials Science, vol 14. Springer, Berlin, Heidelberg.

[3] Smith et al., J. Phys. Chem. C 2017, 121, 12505-12523

We agree with the Reviewer. Theoretically, one would expect to see many more C_xLiC_x configurations, including AA C32. However, only those 4 stages have experimentally been observed unambiguously and reproducibly. We attribute this behavior to relatively small energy barriers between different Li configurations and fast diffusion, which allows only the most stable configurations to show up experimentally. Measurements at shorter time scales and possibly lower temperatures might reveal the AA C32 and other phase, but this is beyond our present capabilities. We have explained this in the revised manuscript.

Staging usually refers to the formation of staged structure, which are defined as "periodic sequences of host and guest layers, with n layers of the host material separating neighboring guest layers in a stage n intercalation compound. In some cases, the staged structures are three-dimensionally ordered crystals, but very often this is not the case. Thus staging is commonly regarded as being a quasi-one-dimensional ordering phenomenon." [REF 2]. The fact that the authors refer to lithium ordering in the plane between the two adjacent graphene layers as "intercalation stages" is somehow confusing, as there is no "staging" phenomena, but different lithium ordering.

We acknowledge the potential for confusion because of the established terminology for intercalated graphite. To address this, we have introduced the term 'in-plane staging' to distinguish it from the traditional concept of staging in graphite, which involves out-of-plane interactions. We considered many

other alternatives, like 'staged in-plane configurations' and 'Li-ion superlattices', but settled on 'in-plane staging' as a rather concise and clear term. We hope the Reviewer agrees with our choice of words. To avoid any confusion, we have made sure that adjective 'in-plane' is used in front of 'staging' throughout the entire manuscript.

Reply to comments of Reviewer #4:

The current manuscript reports on the mechanism of lithium (Li) ions intercalation in bilayer graphene (BLG) systems. The work shows that the intercalation on BLG occurs via four distinct stages (in-plane stages I to IV), which is different from what occurs on graphite. The work also shows that the maximum Li ion density that can be inserted in BLG systems should be at the order of 2.7×10^{14} , which can be achieved at stage IV and correspond to the stoichiometric compound $C_{14}Li_{14}$. However, an essential step to achieve the highest Li storage for BLG is attributed to AB to AA stacking transition, which is facilitated by the AB/BA boundaries formed during intercalation-deintercalation cycles.

The work was conducted very carefully and the reproducibility of the data is impressive, being observed in many different fabricated devices. The agreement between experiment and the DFT calculations in this work is also very relevant and assisted in providing robustness to the experimental data and the conclusions taken in the work. Hence, I do strongly recommend the paper for publication.

We thank the Reviewer for their favourable assessment of our work.

7

I suggest the following questions points for further elaboration and additional discussion:

1) References 7 and 8 show that the denser arrangements should be expected for Li ions intercalation on BLG, corresponding to the compound C_6Li_6 in contrast to the densest arrangement found in stage IV of this work which correspond to the stoichiometric composition of $C_{14}Li_{14}$. Reference 7 shows compelling evidence via in-situ TEM that even close-packed metallic Li could be intercalated in between the layers and that the conclusions presented in the paper should hold irrespective to the stacking order. Reference 8, in turn, shows that there is no fundamental difference between bilayer or few-layer graphene with respect to Li storage manner and kinetic behavior and that the saturated composition should be C_6Li_6 . The authors should, therefore, clarify these apparent contradictions. Why the configuration that would give the highest Li density is $C_{14}Li_{14}$ instead of C_6Li_6 , such as in the previous reports?

We agree that there are somewhat contradictory statements in the literature regarding the maximum Li ion capacity of bilayer graphene. As noted in the introduction, ref. [7] involved Li ion intercalation under a high-energy electron beam, which might interfere with the electrochemical process by contributing to Li ion reduction and formation of metallic lithium between the layers – a different process from reversible ion intercalation which does not correspond to any of the known or predicted compositions (C_6Li_6 or any other). Ref. 8 studied bilayer graphene ‘foam’ that is a bulk structure consisting predominantly (but not exclusively) of bilayers. It appears that the composition of the lithiated foam was determined using a combination of X-ray photoelectron spectra, XRD and SAED data and DFT calculations. In the end the authors concluded that because ‘there is no fundamental difference between bilayer, few-layer graphene or graphite’, the saturated composition must be the same, i.e. C_6Li_6 . At the same time, the reported specific capacity of the bilayer foam (Fig. 2a in ref. [8]) is ~ 180 mAh g⁻¹, twice lower than the 372 mAh g⁻¹ for saturated LiC_6 intercalated graphite and probably requires further understanding of all the factors involved. We study a different, relatively simple, experimental system – small, perfectly flat, defect-free bilayers in an electrochemical setup – so it is not surprising that their results differ from ours. Our findings are reproduced on different devices (new Fig. S6 in the Supplementary Information). Moreover, the maximum Li density found in our work (2.7×10^{14} cm⁻²) is in agreement with other studies of intercalation of pure isolated bilayers (refs. [5,6,9] in the manuscript). We have briefly referred to refs [7,8] in the introduction, and we hope the Reviewer agrees that going into further details would be inappropriate.

2) Since previous reports have shown that Li ion density achievable in BLG systems is about 3 times lower than the one achieved in bulk graphite and considering that it is hard for BLG to replace graphite in practical applications, the authors should clarify on the motivation and significance of the work.

Our work was initially motivated by the lack of understanding of what determines the limits for Li intercalation in bilayer graphene. Although the answer may seem disappointing for potential applications, having a well-substantiated understanding of the intercalation process is significant. Currently ‘graphene batteries’ use multilayer graphene in one of the electrodes and it is working quite well. It is tempting to use the thinnest possible variant (graphene bilayers) to make use of their higher conductivity, ultrafast Li diffusion and large surface area but our work serves as a warning that these superior characteristics come with the reduced storage capacity.

We identified previously unknown essential characteristics of the intercalation process: that intercalation occurs in AB-stacked bilayer graphene without immediate restacking to AA configuration; restacking to AA requires achieving a finite, rather large, Li ion density and is itself required to achieve saturation in Li content. We have shown that the limitation is due to less efficient screening of interionic interactions (enhanced Li ion repulsion) in the bilayer compared to bulk graphite and identified four stages (ordered Li configurations) underpinning intercalation.

Another surprising finding is the experimental evidence for highly ordered Li configurations (essentially Li ion superlattice), which is of interest for electronic transport properties more generally. It would be

interesting to visualize the suggested C_xLiC_x configurations by other techniques, especially STM. We have added this in our conclusions in the revised manuscript.

3) In the paragraph contained in lines 196 to 209, the authors should explain how they assigned the stoichiometric compositions of $C_{42}LiC_{42}$, $C_{38}LiC_{38}$, $C_{18}LiC_{18}$ and $C_{14}LiC_{14}$ to stages I to IV respectively. The

8 authors should consider, perhaps in the supplementary material, to show in a more careful how to assign the number N of carbon atoms to the Li ion densities measured.

To assign stoichiometric compositions to intercalation stages, we calculated Li:C ratios (N in C_NLiC_N) using the extracted average Li densities n_{Li} : $\delta = 2\delta_{Li} \delta'$, where $\delta' = 5.23 \times 10^{-16} \text{ cm}^2$ is the area of a hexagonal unit cell containing 2 C atoms. The values of N calculated in this way for stages I, II, III and IV were 42.0, 37.8, 17.2, 13.9 which were rounded to the nearest integer yielding $C_{42}LiC_{42}$, $C_{38}LiC_{38}$, $C_{18}LiC_{18}$ and $C_{14}LiC_{14}$ (rounding to 18 for stage III ensured Li positions in the centres of C hexagons). Additionally, we have calculated the expected n_{Li} for $N=42, 38, 18$, and 14 as $\delta_{Li} = 2\delta_{Li} \delta'$ these are shown as horizontal lines in Fig. 3a. There is a good agreement between the predicted and measured n_{Li} for each cycle, not just with the average values. We thank the Reviewer for raising this question. This is now explained in the revised manuscript.

Moreover, it would be highly valuable and it would make the results more robust if the authors could provide experimental validation of the stoichiometries of the lithiated products, such as $C_{14}LiC_{14}$, via Raman, XPS, EELS or any other experimental data or at least comment on feasibility.

As detailed in our reply to reviewer #2, the revised manuscript provides an alternative experimental validation of the intercalation process using in operando Raman spectroscopy. The results are shown in new Fig. 5. It clearly demonstrates a large, abrupt change in the electron doping of BLG concurrent with the change in the device resistance and provides an independent experimental validation of intercalation. See our reply to reviewer #2 for further details. Following the Reviewer's comment, the revised manuscript also emphasizes that it would be desirable to visualize the suggested C_xLiC_x configurations by other techniques, especially STM.

4) The manuscript highlights the difference in lithium ion behavior between bilayer graphene (BLG) and bulk graphite, attributing it to the weaker screening effect in thin layers. However, in reference 35, which was one of the references cited in the work to back up this argument, the authors even with the reduced screening potential, the most stable Li density should be approximately LiC_7 , which is different from the LiC_{14} found in this work. The authors should comment on this.

Moreover, the screening argument raises a question about the transition from 2D to bulk behaviour: At what point, in terms of the number of graphene layers, does this 2D screening effect diminish sufficiently for lithium ions to behave similarly to those in bulk graphite, without significant repulsion? Clarifying this aspect would provide valuable insight into how the properties of graphene-based materials evolve from the nanoscale to the bulk scale. In reference 34, it seems that the screening effect changes significantly from the monolayer to bilayer and to the rest (few-layer and bulk), but the screening in the bilayer does not significantly changes compared to few-layered or bulk systems?

We apologise for not being clear and for the resulting confusion. Ref. 34 is cited as an indication of the importance of dimensionality for screening effects, although it considers dielectric rather than metallic screening. The screening which matters for intercalated Li ions is screening of the ionic repulsion by electrons transferred to BLG as the result of Li intercalation.

The importance of electrostatic repulsion between intercalated ions in layered materials, where ions intercalate into interlayer galleries, has long been appreciated and discussed most notably for graphite (refs [1-4] below) and other crystals, such as MXenes and TiS_2 (refs [5,6] below). The authors of refs 38,41 in the revised manuscript used a simple model of point charges interacting through screened electrostatic interactions to argue that screening naturally has a strong effect on attainable intercalation densities. This effect is especially important for intercalation of ions into BLG, where the same planar density of Li ions per carbon layer (C_xLiC_x) as in bulk graphite (LiC_x) results in less Li-dense configurations overall simply because the same number of Li ions are shared by the two adjacent layers in BLG but not in bulk graphite. As the result, the host carbon atoms are less strongly doped, because the amount of charge lost by one Li ion to the surrounding graphene layers is approximately the same in both cases (ref. [11] in the revised manuscript). Accordingly, the density of charge carriers at the Fermi level in Li-intercalated BLG is lower, which in turn leads to relatively ineffective screening of the interionic repulsion.

9

Our DFT calculations naturally incorporate the effect of screening and allow numerical estimation of its impact. We find a very strong effect of the poorly screened Coulomb repulsion on ionic mobility, which is shown in new figure (Fig. S11) through comparison of off-stoichiometric arrangements of Li ions in a repeated $7 \text{ \AA} \times 7 \text{ \AA}$ superlattice. It demonstrates a significantly higher energetic penalty in the bilayer case, in agreement with the qualitative arguments above. These physical considerations set the conditions for the lower attainable Li density and in-plane staging experimentally observed in this work.

We have added the above discussion in the revised manuscript.

We agree with the Reviewer that layer dependence of this screening is another important question.

However, quantifying further the exact effect of screening vs the number of layers would require substantial numerical effort and taking into account other important factors, such as interlayer staging and long-range strain between nucleated domains. These must be considered for trilayer and tetralayer graphene and

should be a subject of further research.

5) Finally, the authors should comment why the Li ions are expected to predominantly intercalate within the interlayer of BLG and why surface placement on the bilayer can be neglected.

The fact that Li ions do not adsorb on the surface of graphene and do not intercalate between e.g. monolayer graphene and Si/SiO₂ substrate, has already been established in literature both theoretically (ref. [11] in the revised manuscript) and experimentally (ref. [9]). Our control experiments using monolayer graphene confirmed no intercalation between graphene and SU-8 or Si/SiO_x substrates. Intercalation only occurred in the bilayer gallery, in agreement with previous studies. We noted this in the revised manuscript and thank the Reviewer for raising this interesting question.

1. Allart et al, Model of lithium intercalation into graphite by potentiometric analysis with equilibrium and entropy change curves of graphite electrode. *J. Electrochem. Soc.* 165, A380 (2018).

2. Annis, S. et al. Accessing structural, electronic, transport and mesoscale properties of Li-GICs via a complete DFTB model with machine-learned repulsion potential. *Materials* 14, 6633 (2021).

3. Panosetti et al. DFTB modeling of lithium-intercalated graphite with machine-learned repulsive potential. *J. Phys. Chem. A* 125, 6916-6999 (2021).

4. Pande. V. and Viswanathan, V. Robust high-fidelity DFT study of the lithium-graphite phase diagram. *Phys. Rev. Materials* 2, 125401 (2018).

5. Eames, C. & Saiful Islam, M. Ion intercalation into two-dimensional transition-metal carbides: Global screening for new high-capacity battery materials. *J. Am. Chem. Soc.* 136, 16270-16276 (2014).

6. Jacobsen, T., West, K., Atlung, S. Electrostatic interactions during the intercalation of Li in Li_xTiS₂. *Electrochimica Acta* 27, 1007-1011 (1982).

10

Version 1:

Reviewer comments:

Reviewer #1

(Remarks to the Author)

The revised manuscript includes Raman spectroscopy characterization of lithium intercalation in the BLG device, but there is still a lack of the experimental evidence for the transition from AB to AA stacking. To achieve a Raman spectrum with good quality, Raman measurements can be conducted in an inert atmosphere protected by electrochemical-Raman cell (see publications in Ref. [1-4]) without using SU8. Alternatively, the stacking changes between AB and AA in a bilayer graphene can be observed by XRD according to the previous study Ref [1].

Despite this, it does not affect the overall innovativeness of the article in utilizing the magnetotransport measurements to study the in-plane ordering of lithium ions inside the BLG. More importantly, we are aware of how challenging the device fabrication and measurement are.

Overall, the author has responded reasonably to the questions raised by the Reviewers, addressing the doubts regarding the experimental details and providing a very detailed explanation of the electrochemical testing conditions and device specifics. We can recommend the publication of this manuscript.

[1] Lithium intercalation into bilayer graphene, *Nature Communications*.

[2] Operando nano-mapping of sodium-diglyme co-intercalation and SEI formation in sodium ion batteries graphene anodes, *Applied Physics Reviews*.

[2] Nanoarchitecture factors of solid electrolyte interphase formation via 3D nano-rheology microscopy and surface force-distance spectroscopy, *Nature Communications*.

[4] In Operando Probing of Lithium-Ion Storage on Single-Layer Graphene, *Advanced Materials*.

Reviewer #2

(Remarks to the Author)

I am satisfied with the revisions implemented by the authors. Happy to read that my suggestions allowed to strengthen the manuscript.

Reviewer #3

(Remarks to the Author)

The authors make a great job to clarify the questions raised by the different reviewers and conducted additional experiments (Raman) to consolidate their interpretation. Especially, they added clarifications regarding the electrochemical set-up and the source of lithium (the solid electrolyte).

The only concern that I still have regards the time evolution of the lithium intercalated/de-intercalated into the BLG and the state of lithiation of the BLG during the plateaus of ρ_{xx} . The authors suggest that a plateau corresponds to one state of lithiation with a given concentration (stage I for example). As suggested by reviewer #1, the state of lithiation can be estimated by monitoring the cathodic/anodic current between the BLG and the counter electrode. The current during the intercalation is presented in the revised version on the bottom of FigS7a. The evolution of the intercalated lithium in the BLG should be the integral of this current over time. This would lead to a continuous increase (even if with different slopes) of lithium in the BLG, in contradiction with the interpretation proposed by the author suggesting a state of lithiation of the BLG with constant values during the plateau of ρ_{xx} and jumps from one state of lithiation to the next one. This later estimation of the intercalated lithium, is based on the interpretation of graphene's resistivity and Hall Voltage's measurements, techniques for which I am not a specialist. Therefore, I do not know if the growth of domains with higher lithium densities should lead to a smooth increase of ρ_{xx} (as suggested by the authors line 241), or if ρ_{xx} can be related to other physical quantities as the chemical potential which stays constant during domains growth of phase co-existences.

Apart from this point, I think that the authors did a significant experimental and computational work, which brings new elements to be discussed by the community. Therefore, I think that this paper deserves publication.

Reviewer #4

(Remarks to the Author)

I am satisfied with the changes made and the MS can be accepted as is.

Author Rebuttal letter:

Reviewer #1:

The revised manuscript includes Raman spectroscopy characterization of lithium intercalation in the BLG device, but there is still a lack of the experimental evidence for the transition from AB to AA stacking. To achieve a Raman spectrum with good quality, Raman measurements can be conducted in an inert atmosphere protected by electrochemical-Raman cell (see publications in Ref. [1-4]) without using SU8. Alternatively, the stacking changes between AB and AA in a bilayer graphene can be observed by XRD according to the previous study Ref [1].

Despite this, it does not affect the overall innovativeness of the article in utilizing the magnetotransport measurements to study the in-plane ordering of lithium ions inside the BLG. More importantly, we are aware of how challenging the device fabrication and measurement are.

Overall, the author has responded reasonably to the questions raised by the Reviewers, addressing the doubts regarding the experimental details and providing a very detailed explanation of the electrochemical testing conditions and device specifics. We can recommend the publication of this manuscript.

[1] Lithium intercalation into bilayer graphene, Nature Communications.

[2] Operando nano-mapping of sodium-diglyme co-intercalation and SEI formation in sodium ion batteries graphene anodes, Applied Physics Reviews.

[2] Nanoarchitecture factors of solid electrolyte interphase formation via 3D nano-rheology microscopy and surface force-distance spectroscopy, Nature Communications.

[4] In Operando Probing of Lithium-Ion Storage on Single-Layer Graphene, Advanced Materials.

We thank the Reviewer for his/her favorable assessment of the revised manuscript and for recommending it for publication.

We are also grateful for pointing out the additional references, particularly the detailed studies of the solid-electrolyte interphase formation by using Raman spectroscopy. We have added those references to the final version of the manuscript: new refs. [33,34] in the main text and new ref. [2] in the Supplementary Information. The Reviewer's ref. 1 above has been included already (ref. [8] in the main text). As suggested by the Reviewer, our Raman data were obtained using an inert atmosphere within an electrochemical cell of a similar design to that used in [1-4] but with additional electrical connections that allowed Raman measurements to be carried out simultaneously with the electrical ones. This was essential to correlate the observed changes in transport properties to those in Raman spectra. Otherwise, we would not be able to assign the changes in Raman spectra to stages of intercalation for our bilayer graphene. For this reason, we still needed to use the SU8 protection so that the sensitive part of the device was separated from the electrolyte to prevent any spurious ionic gating.

We appreciate the Reviewer's suggestion to use XRD to detect changes from AB to AA stacking, as had been used for bulk samples in the past, including Ref. [1] on the Reviewer's list. In our future studies we plan to use micro-focused X-rays similar to the method used in Ref. 12 of our manuscript. The authors of Ref. [12] were able to detect a subtle expansion of few-layer graphene crystals under lithiation and the method may provide sufficient sensitivity to detect other structural changes. At present, we do not have access to this specialized equipment required for studies of our micrometer-sized samples.

Reviewer #3 (Remarks to the Author):

The authors make a great job to clarify the questions raised by the different reviewers and conducted additional experiments (Raman) to consolidate their interpretation. Especially, they added clarifications regarding the electrochemical set-up and the source of lithium (the solid electrolyte).

The only concern that I still have regards the time evolution of the lithium intercalated/de-intercalated into the BLG and the state of lithiation of the BLG during the plateaus of ρ_{xx} . The authors suggest that a plateau corresponds to one state of lithiation with a given concentration (stage I for example). As suggested by reviewer #1, the state of lithiation can be estimated by monitoring the cathodic/anodic current between the BLG and the counter electrode. The current during the intercalation is presented in the revised version on the bottom of FigS7a. The evolution of the intercalated lithium in the BLG should be the integral of this current over time. This would lead to a continuous increase (even if with different slopes) of lithium in the BLG, in contradiction with the interpretation proposed by the author suggesting a state of lithiation of the BLG with constant values during the plateau of ρ_{xx} and jumps from one state of lithiation to the next one. This later estimation of the intercalated lithium, is based on the interpretation of graphene's resistivity and Hall Voltage's measurements, techniques for which I am not a specialist. Therefore, I do not know if the growth of domains with higher lithium densities should lead to a smooth increase of ρ_{xx} (as suggested by the authors line 241), or if ρ_{xx} can be related to other physical quantities as the chemical potential which stays constant during domains growth of phase co-existences.

Apart from this point, I think that the authors did a significant experimental and computational work, which brings new elements to be discussed by the community. Therefore, I think that this paper deserves publication.

We are grateful to the Reviewer for this favorable assessment of our work and for recommending the revised manuscript for publication.

We agree that the measured cathodic-anodic currents during intercalation/deintercalation provide a measure of the amount of lithium entering/leaving the device. Following earlier suggestion of Reviewer #1 and, also, as suggested by the Reviewer here, we used the integral of the step-like change in the cathodic current in Supplementary Fig. 7b, corresponding to the resistance step from stage I to stage III, to estimate the amount of lithium entering the device and found good agreement with the value extracted from the electrical measurements (changes in graphene's carrier density). Regarding the possibility of coexisting domains of different densities as observed in intercalated graphite, this scenario certainly contradicts our observations. As now emphasized in the final manuscript, a gradual increase of the density of lithium (for example, if a higher density Li domain were growing at the expense of a lower density one) would be seen as a gradual increase in the amount of charge transfer from Li to graphene and, therefore, of the measured carrier density (using the Hall effect). The absence of such behavior unequivocally allowed us to conclude that the observed plateaus in longitudinal and Hall resistances correspond to constant Li densities. In future studies, we aim to complement magnetotransport measurements with other methods, including more accurate measurements of cathodic-anodic currents, micro-probe XRD and possibly STM/AFM as suggested by the Reviewers.
